



# A comparison of Eulerian and Lagrangian methods for vertical particle transport in the water column

Tor Nordam[1,2], Ruben Kristiansen[2], Raymond Nepstad[1], Erik van Sebille[3], and Andy M. Booth[1]

[1]SINTEF Ocean AS, Trondheim, Norway
[2]Department of Physics, Norwegian University of Science and Technology, Trondheim, Norway
[3]Institute for Marine and Atmospheric Research, Utrecht University, Utrecht, The Netherlands

**Correspondence:** Tor Nordam (tor.nordam@sintef.no)

**Abstract.** A common task in oceanography is to model the vertical movement of particles such as microplastics, nanoparticles, mineral particles, gas bubbles, oil droplets, fish eggs, plankton, or algae. In some cases, the distribution of vertical rise or settling velocities of the particles in question can span a wide range, covering several orders of magnitude, often due to a broad particle size distribution or differences in density. This requires numerical methods that are able to adequately resolve a wide and possibly multi-modal velocity distribution.

Lagrangian particle methods are commonly used for these applications. A strength of such methods is that each particle can have its own rise or settling speed, which makes it easy to achieve a good representation of a continuous distribution of speeds. An alternative approach is to use Eulerian methods, where the partial differential equations describing the transport problem are solved directly with numerical methods. In Eulerian methods, different rise or settling speeds must be represented as discrete classes, and in practice only a limited number of classes can be included.

Here, we consider three different examples of applications for a water-column model: positively buoyant fish eggs, a mixture of positively and negatively buoyant microplastics, and positively buoyant oil droplets being entrained by waves. For each of the three cases we formulate a model for the vertical transport, based on the advection-diffusion equation with suitable boundary conditions and in one case a reaction term. We give a detailed description of an Eulerian and a Lagrangian implementation of these models, and we demonstrate that they give equivalent results for selected example cases. We also pay special attention to the convergence of the model results with increasing number of classes in the Eulerian scheme, and the number of particles in the Lagrangian scheme. For the Lagrangian scheme, we see the $1/\sqrt{N_p}$ convergence as expected for a Monte Carlo method, while for the Eulerian implementation, we see a second order $(1/N_k^2)$ convergence with the number of classes.

## 1 Introduction

Studying the vertical transport of positively, negatively, or neutrally buoyant particles is a common task in oceanography. Examples include both anthropogenic and naturally occurring particles, such as microplastics, mineral particles, nanoparticles, aggregates, gas bubbles, oil droplets, fish eggs, or even particles with active swimming behaviour such as zooplankton. These particles may show a range of different behaviours, including rising, sinking, and interacting with the ocean surface and sea floor in different ways.





Vertical transport modelling may be applied at different scales. In a simple one-dimensional water-column model the goal may be to investigate the timescale of settling or surfacing for a specific type of particle. However, accurate modelling of vertical transport is also key to predicting horizontal transport at largel scales, due for example to the vertical variability of horizontal ocean currents (Röhrs et al., 2018; Wichmann et al., 2019). Hence, a good description of vertical behaviour is an essential part of any three-dimensional model.

Commonly used transport models may be divided into two classes: Eulerian and Lagrangian models. Eulerian models consist of solving the advection-diffusion-reaction equation directly with numerical methods for partial differential equations. Ocean circulation models, such as ROMS (Shchepetkin and McWilliams, 2005) or NEMO (Gurvan et al., 2022), are typical examples of Eulerian models. Related are Computational Fluid Dynamics (CFD) models, which can be used, for example, to model waves and turbulence on smaller scales (Cui et al., 2020). Eulerian models are also used to compute the transport of

suspended particles in water, for example natural sediments (Warner et al., 2008), nanoparticles in rivers (Saharia et al., 2019), or microplastics in the oceans (Mountford and Morales Maqueda, 2019).

  A challenge of the Eulerian approach, when it comes to particles with a distribution of rise or settling speeds, is that one must use discrete speed classes, and solve the advection-diffusion equation for each class, forming a large system of equations to solve simultaneously.

Lagrangian particle methods are quite a popular choice for modelling particle transport in the ocean (van Sebille et al., 2018). In these methods, numerical particles, also called Lagrangian elements, are used to represent physical particles. The numerical particles will move with the current, may exhibit some form of random displacement to model turbulent diffusion, and may have a vertical rise or settling speed (or even active swimming behaviour). Lagrangian particle methods have been applied to transport modelling of a wide range of particle types and processes, including plastics (Delandmeter and Van Sebille,

2019; De La Fuente et al., 2021; Fischer et al., 2022), residence time of water masses (Dugstad et al., 2019), transport and sedimentation of mineral particles in the ocean (Dissanayake et al., 2014; Nepstad et al., 2020), surfacing and entrainment of oil (Cui et al., 2018; Nordam et al., 2019b), transport of dissolved gases (Dissanayake et al., 2012; Wimalaratne et al., 2015), produced water (Nepstad et al., 2022), harmful algae (Rowe et al., 2016), dinocysts and foraminifera (Van Sebille et al., 2015; Nooteboom et al., 2019), fish eggs (Sundby, 1983; Röhrs et al., 2014), and even fish (Scutt Phillips et al., 2018). Stochastic

particle transport methods are also used in many other fields of science, including the study of cosmic rays and high-energy particles from the sun (Strauss and Effenberger, 2017), deposition of inhaled nanoparticles in the airways (Longest and Xi, 2007), and transport modelling of airborne virus transmission (Abuhegazy et al., 2020).

  One of the advantages of a Lagrangian approach to particle transport modelling is the ability to represent a wide range of properties or behaviours. By letting each numerical particle with its own properties move independently of the others, one can

model physical particle distributions where the sizes, and hence terminal velocities, can vary by several orders of magnitude.

  The purpose of this paper is to compare and discuss Eulerian and Lagrangian methods, with a focus on the numerical implementation of the models, including different boundary conditions and a reaction term. We demonstrate that the two implementations give the same results, and we also address questions of efficiency and convergence. The methods are illustrated using three different one-dimensional cases as a basis for the discussion: Fish eggs, microplastics and oil droplets. The cases





have been chosen because they represent simplified (but realistic) cases, and demonstrate how different boundary conditions and a simple reaction term may be implemented in both Eulerian and Lagrangian schemes.

In this paper, we consider the water column without background flow, and investigate the vertical transport of different types of particles. In Section 2, we describe the advection-diffusion equation, which is a partial differential equation (PDE) that will form the basis of our transport problems. Additionally, we present a stochastic differential equation (SDE), which yields a Lagrangian particle method that is equivalent to the advection-diffusion equation. In Section 3, we briefly outline how we implement an Eulerian model, based on numerically solving the advection-diffusion equation with PDE methods, and similarly how we implement a Lagrangian model, based on numerically solving the equivalent SDE for a large ensemble of particles.

In Section 4, we apply our Eulerian and Lagrangian water-column models to three different cases: Fish eggs, microplastics, and oil droplets. All three cases are modelled both with the Eulerian approach and the Lagrangian approach. In each case, the particles considered have a wide range of rising or sinking speeds. Additionally, the cases serve to highlight the effects of different boundary conditions, and a reaction term. In Section 5 we discuss and summarise our comparison of the Eulerian and Lagrangian approaches. Finally, in Section 6, we present some concluding remarks.

Some additional details are given in the Appendices: Appendix A contains detailed descriptions of our Eulerian and Lagrangian model implementations, Appendix B has details on how we obtained a distribution of terminal velocities for microplastics, and Appendix C shows additional numerical convergence results.

## 2 Components of the advection-diffusion reaction model

### 2.1 The advection-diffusion equation

As our starting point, we assume that the movement of a collection of particles with different rise or settling velocities in the water column may be described by the advection-diffusion equation (see, *e.g.*, Hundsdorfer and Verwer (2003)). If $C_k(z,t)$ is the concentration of particles with (constant) vertical rise or settling speed $w_k$, then we have

$$\frac{\partial C_k}{\partial t} = \frac{\partial}{\partial z}\left(K(z)\frac{\partial C_k}{\partial z}\right) - \frac{\partial}{\partial z}\big(w_k C_k\big), \tag{1}$$

where $K(z)$ is the diffusivity. To represent a distribution of particle sizes, densities, and/or shapes, and thus a distribution of rise speeds, one must in principle solve a (coupled) set of advection-diffusion equations, with one equation for each particle speed, $w_k$.

### 2.2 Equation of motion of Lagrangian particles

A mathematically equivalent formulation of the advection-diffusion problem is to consider an ensemble of numerical "particles", whose positions develop in time according to the stochastic differential equation (SDE)

$$\mathrm{d}z = \big(w + K'(z)\big)\mathrm{d}t + \sqrt{2K(z)}\,\mathrm{d}W_t. \tag{2}$$





Here, $W_t$ is a standard Wiener process (Kloeden and Platen, 1992, p. 40), $K(z)$ is again the diffusivity as a function of depth, $K'(z)$ is its derivative with respect to $z$, and $w$ is the terminal velocity due to buoyancy, which is constant for a given particle.

The connection between Eqs. (1) and (2) is that the probability distribution for the position of particles moving according to Eq. (2), will develop according to Eq. (1). A solution, $z(t)$, to Eq. (2) is called an Itô-diffusion and Eq. (1) is the corresponding Fokker-Planck equation (Kloeden and Platen (1992, p. 37), see also Appendix A of Nordam et al. (2019b)). Each numerical particle position, determined from the SDE, represents a sample from the probability distribution of particle positions, which develops according to the PDE. Solving Eq. (2) for a sufficient number of particles allows us to approximate the probability distribution, which is proportional to the concentration in the Eulerian scheme.

## 2.3 Velocity distribution

In many practical applications, one will have a size distribution of particles, obtained from measurements, a model, or other estimates. For the model we consider here, however, the relevant property is not the size itself, but the terminal rise or settling speed of the particle, which is a function of size (and other particle parameters such as shape and density, as well as the viscosity of the ambient fluid). Note that we assume here that a particle will immediately attain its terminal velocity, and that particle motion can be described as the combination of a constant terminal velocity and series of random displacements. This is a commonly used approximation, which holds well if the timescale needed to reach terminal velocity is short compared to other relevant timescales. As an example, consider a particle with some initial speed $v_0$, moving in water under the influence of Stokes' drag (see Appendix B). Then we have that the initial speed will decay exponentially, as given by

$$v(t) = v_0 e^{-t/\tau}, \tag{3}$$

where $\tau = \frac{m}{6\pi\mu r}$, with $\mu$ being the dynamic viscosity of water, $r$ the radius of the particle and $m$ the mass of the particle. For typical values for small particles in water, the timescale $\tau$ is less than a second (*e.g.*, $\mu = 1.7 \times 10^{-3}\,\mathrm{Pa\,s}$, $r = 1 \times 10^{-3}\,\mathrm{m}$, $m = 4 \times 10^{-6}\,\mathrm{kg}$ gives $\tau = 0.14\,\mathrm{s}$).

In the example cases we consider in Section 4, we will assume that we have access to the distribution of terminal velocities, either by directly specifying a functional form of the velocity distribution, or by numerical means through mapping particle size distribution and other properties to velocities.

## 2.4 Boundary conditions

Depending on the application, different boundary conditions may be used to control the fluxes across the domain boundaries in a water-column model. We will deal separately with the diffusive flux,

$$j_D(z) = -K(z) \left.\frac{\partial C_k}{\partial z}\right|_z, \tag{4}$$

and the advective flux

$$j_A(z) = w_k C_k(z). \tag{5}$$





For the boundary conditions for the diffusive flux, we first observe that diffusion in our model represents the mixing that
occurs due to the combination of random turbulent fluctuations and molecular diffusion (Thorpe, 2005, p. 20). Being caused
by the motion of the water, the diffusive flux should not allow suspended particles to leave the water column. Hence, we wish
to enforce zero diffusive flux, $j_D = 0$, across the boundaries both at the surface and at the sea floor. This applies in all the cases
we consider in this paper.

Next, we consider the advective flux. In our model, the advection velocity, $w_k$, represents the terminal rising or sinking
velocity of a particle due to buoyancy. Depending on the application, this velocity may or may not allow particles to leave the
water column. As an example, consider positively buoyant fish eggs. When fish eggs rise to the surface, they can of course
rise no further. However, they remain submerged, having a hydrophilic surface and a density only slightly less than that of the
surrounding seawater. Hence, we have chosen to model fish eggs with a zero-flux boundary condition at the surface. With the
surface at $z = 0$ (and $z$ negative downwards), the total flux, $j_{tot}(z)$, at the surface is then given by

$$j_{tot}(0) = j_A(0) + j_D(0) = 0. \tag{6}$$

As another example, consider positively buoyant oil droplets. When oil droplets rise to the surface, they go from being
individual droplets surrounded by water, to forming small patches of floating oil at the surface or possibly merging with a
larger surface slick. When this happens, the oil droplets are no longer subject to random motion due to turbulent diffusion, and
the oil will remain at the surface until some high-energy event like a breaking wave causes the surface slick to break up and
form droplets.

To express the mechanism of oil droplets surfacing and leaving the water column, we have two options: We can use a loss
term, which removes oil droplets close to the surface at some rate (Tkalich and Chan, 2002), or we can use a partially absorbing
boundary condition. Here, we have chosen to allow the advective flux to carry oil droplets across the boundary at the air-sea
interface, while still enforcing zero diffusive flux across the surface:

$$j_{tot}(0) = j_A(0) + j_D(0) = w_k C_k(0). \tag{7}$$

The reasons we describe surfacing of oil droplets through the boundary condition instead of a loss term are that it is straightforward to express the boundary condition both in the Eulerian and the Lagrangian implementations, and that we do not have to
define what it means to be "close to the surface" for the loss term. The idea behind allowing an advective flux, while forcing the
diffusive flux to be zero, is that buoyancy is the mechanism that leads to surfacing. Additionally, if the diffusive flux out of the
system is nonzero, then increased diffusivity would lead to faster surfacing, which is contrary to observations. For additional
discussion of this point, see Nordam et al. (2019a).

We note that similar reasoning as above applies also to the boundary at the sea floor. For example, the settling out of
negatively buoyant particles (microplastics, sediments, etc.) can be modelled as an advective flux leaving the model domain
through the bottom boundary (*i.e.*, settling on the seafloor or incorporation into sediments). Note also that "advective flux" here
means the flux due to the rise or settling speed of the particle, not due to vertical currents.



## 2.5 Reaction terms

An additional reaction term $S_k(z, C_1, C_2, \ldots, C_{N_k})$ may be included in Eq. (1), in which case it becomes the advection-diffusion-reaction equation. It can describe the addition of mass (a source), the removal of mass (a sink), or the transformation of mass between classes, for example from $C_1$ to $C_2$, which means that in general the reaction term for class $k$, can depend

on the concentration of all classes. In an Eulerian implementation, a reaction term can simply add or remove mass in different locations. This is described in Section A1.4. In a Lagrangian stochastic particle implementation, Eq. (2) only describes the *transport* of numerical particles, and reaction terms have to be implemented as a separate step, with details depending on the nature of the reaction. This is discussed in more detail in Section A2.2.

For one of the cases considered in Section 4, we will use a source term to describe the addition of mass in a limited region

of the computational domain. This will be used to represent the entrainment of oil droplets from the surface and into the water column.

## 2.6 Sampling error in stochastic particle methods

When we solve an advection-diffusion-reaction problem by means of a stochastic Lagrangian particle model, the position (and possibly other properties) of each numerical particle represents a sample from an underlying distribution. When we draw

random samples from a distribution, and calculate for example a mean of the samples, there will be a random error in the sample mean, relative to the true (but usually unknown) mean of the distribution. In particular, if we draw $N_p$ independent samples from a distribution with finite variance, and take the mean of those samples, this is called the sample mean, $\mu_{N_p}$, which is an approximation of the true mean, $\mu$. By the strong law of large numbers, we have that $\mu_{N_p} \to \mu$ as $N_p \to \infty$, almost surely (Billingsley, 1979, p. 85). Furthermore, according to the Lindeberg-Lévy theorem (Billingsley, 1979, p. 308), we have

$$\sqrt{\frac{N_p}{\sigma^2}} |\mu_{N_p} - \mu| \sim \mathcal{N}(0, 1), \tag{8}$$

where $\sigma^2$ is the (assumed finite) variance of the distribution from which the samples are drawn, and $\mathcal{N}(0, 1)$ is a standard normal distribution with zero mean and unit variance. In other words, in an ensemble of simulations the absolute error in the sample mean, $|\mu_{N_p} - \mu|$, will be a Gaussian random number, with standard deviation $\sqrt{\sigma^2/N_p}$.

## 3 Implementation

### 3.1 Eulerian model

There is a wide range of numerical methods for PDEs to choose from for solving the advection-diffusion-reaction equation (see for example Hundsdorfer and Verwer (2003)). Here, we have chosen to use a finite-volume method (FVM) for discretisation in space, and the Crank-Nicolson (implicit trapezoidal) scheme for discretisation in time. An FVM was selected due to the ease of applying prescribed-flux boundary conditions, and due to the inherent conservation of mass. Our chosen approach leads to





second-order convergence in both space and time, and the Crank-Nicolson scheme is also known to be unconditionally stable for the diffusion equation (Versteeg and Malalasekera, 2007, p. 247).

    A detailed description of the discretisation, as well as the boundary conditions and the numerical solver, is given in Appendix A. Below, we focus on the three most relevant implementation details for the later discussion: A brief review of the boundary conditions, discretisation of the velocity distribution into classes, and a measure of the numerical error.

### 185   3.1.1   Boundary conditions

As mentioned in Section 2.4, we wish to use two different types of boundary conditions: zero-flux and zero-diffusive-flux. Using a finite-volume method, our Eulerian implementation represents space as a grid of cells, with the changing concentration in a cell expressed in terms of the sum of the fluxes across the cell faces. The fluxes are approximated numerically from the concentrations in the neighbouring cells, as well as the diffusivity and the advection velocity for the diffusive and advective

fluxes respectively. Details are given in Appendix A1.2.

    With a finite-volume method, it is trivial to implement prescribed-flux boundary conditions by simply setting one or both of the advective and diffusive fluxes across the cell face at the end of the domain to the prescribed value. In our case, we implement no-flux boundary conditions by setting both fluxes to zero across the boundary, and no-diffusive-flux by setting only the diffusive flux to zero, leaving the advective flux unchanged.

### 195   3.1.2   Discrete representation of velocity distribution

In this study, we are interested in the scenario where our particles have a distribution of rising and/or settling velocities. To investigate the convergence of our Eulerian method with an increasing number of velocity classes, we need an automated way to represent the velocity distribution as a given number of discrete classes. Depending on the application, different ways of dividing the velocity distribution into intervals may be preferable.

In our selected approach, we first choose lower and upper limits; $w_{min}$ and $w_{max}$. We then divide the interval between the limits into $N_k$ classes, with equal spacing on either a linear or logarithmic scale, depending on the case. The representative velocity for each class is chosen to be the midpoint of the class, on linear or logarithmic scales, respectively. For cases where the relevant velocities span several orders of magnitude, a logarithmic spacing may be preferable. Where both negative (sinking) and positive (rising) velocities are represented, they are handled separately if a logarithmic spacing is chosen. In Section 4 we

will describe in detail how the velocity distribution is represented for each of the considered cases.

    To set up the initial condition, the total mass must be divided among the different velocity classes. In some cases, the velocity distribution might be available directly. In other cases, it needs to be inferred from another distribution, most commonly the size distribution combined with particle density, and, in some cases, other particle properties such as shape.

    To set up a case with $N_k$ velocity classes and constant linear spacing, we use a constant class width $\Delta w$ such that $w_k =$

$w_1 + (k-1)\Delta w$. All particles with velocities in the interval $[w_k - \Delta v/2, w_k + \Delta w/2)$ belong to class $k$, and are represented





by the velocity $w_k$ in the Eulerian implementation. The fraction of the total mass belonging to this class is given by

$$\int\limits_{w_k-\Delta w/2}^{w_k+\Delta w/2} p(w)\,\mathrm{d}w, \tag{9}$$

where $p(w)$ is the velocity distribution, normalised such that

$$\int\limits_{w_{min}}^{w_{max}} p(w)\,\mathrm{d}w = 1. \tag{10}$$

Note that $w_{min} = w_1 - \Delta w/2$, and $w_{max} = w_{N_k} + \Delta w/2$.

With equal spacing on a logarithmic scale, the calculation of mass fractions is similar, but the limits on the integral are different. For constant logarithmic spacing, $\delta w$, we have that $\frac{w_{k+1}}{w_k} = \delta w$ for $k = 1, \ldots, N_k - 1$, and the particles belonging to class $k$ are those with velocities in the interval $\left[w_k/\sqrt{\delta w}, w_k \cdot \sqrt{\delta w}\right)$. For completeness, we note that in this case, $w_{min} = w_1/\sqrt{\delta w}$, and $w_{max} = w_{N_k} \cdot \sqrt{\delta w}$.

### 3.1.3 Measures of the numerical error

To compare our Eulerian and Lagrangian implementations, we will present direct comparisons of the predicted concentration, while an investigation of the convergence of our solutions as functions of the different numerical parameters requires a quantitative measure of the error. We have chosen to use the first moment (*i.e.*, the center of mass) of the distribution. The reason for this choice is primarily due to numerical methods for SDEs usually having a well-defined weak convergence in terms of the

moments of the distribution (Kloeden and Platen, 1992, p. 327), making this a natural parameter to consider for the Lagrangian method. We chose to use the same metric for the Eulerian method to facilitate a direct comparison. When used to consider timescales for surfacing or settling of particles, one could also argue that the depth of the center of mass of a distribution is a relevant output from a water column model.

The first moment of a distribution $p(z)$, on the interval $z \in [0, H]$, is given by

$$M_1 = \int\limits_0^H z\,p(z)\,\mathrm{d}z, \tag{11}$$

where the distribution is normalised such that $\int_0^H p(z)\,\mathrm{d}z = 1$.

Numerically, we approximate the first moment for the Eulerian results by calculating the integral in Eq. (11) by the standard midpoint Riemann sum quadrature method (Wendroff, 1969, p. 23), and taking the sum over all velocity classes:

$$M_1(N_k) = \frac{\sum_{k=1}^{N_k}\left(\Delta z \sum_{j=1}^{N_z} z_n C_{k,j}\right)}{\sum_{k=1}^{N_k}\left(\Delta z \sum_{j=1}^{N_z} C_{k,j}\right)}, \tag{12}$$

where $\Delta z$ is the cell size in the spatial discretisation, $C_{k,j}$ is the concentration of particles with velocity $w_k$ in cell $j$, and $z_j$ is the midpoint of cell $j$ (see Fig. A1 and Section A1.1 in the Appendix for description of the spatial discretisation).





As we do not have analytical solutions for the cases considered in Section 4, the convergence analysis is conducted purely with numerical results. To consider the convergence with number of classes, $N_k$, for the Eulerian implementation, we approximate the error in the numerically calculated first moment $M_1(N_k)$ as follows

$$E(N_k) = M_1(N_k) - M_1 \approx M_1(N_k) - M_1(N_k^{\text{ref}}), \tag{13}$$

where $M_1$ is the true (but unknown) first moment of the distribution, and $M_1(N_k^{\text{ref}})$ is a reference solution calculated with a large number of classes. The other numerical parameters (timestep and spatial discretisation) are kept constant, while $N_k$ is varied. As long as $N_k \ll N_k^{\text{ref}}$, we assume that the approximation in Eq. (13) is a good one. See also Appendix C for additional details on this point, including convergence tests used to select suitable values of the timestep and spatial discretisation.

## 3.2 Lagrangian model

The starting point for our Lagrangian implementation for solving the advection-diffusion equation is the SDE given by Eq. (2). We solve this equation numerically for a large ensemble of initial conditions, using the Euler-Maruyama method (Kloeden and Platen, 1992, p. 305). Each solution describes the trajectory of a "numerical particle", and the concentration can be found from the distribution of the numerical particles.

In Appendix A, we provide details of the numerical solution of the SDE, implementation of a reaction term, and the estimation of the concentration field from the ensemble of particles. Below, we describe the implementation of boundary conditions, how the particles are assigned velocities from a distribution, and we describe our measure of numerical error, used later to assess the numerical accuracy.

### 3.2.1 Boundary conditions

As described in Section 2.4, different boundary conditions are relevant for different applications. Here, we wish to employ zero-flux boundary conditions, and zero-diffusive-flux boundary conditions. We also need to implement the same boundary conditions in both the Eulerian and the Lagrangian schemes. The link between the random walk SDE (Eq. (2)) and the advection-diffusion equation (Eq. (1)) through the Fokker-Planck equation is relatively clear and well described in reasonably accessible literature (see Section 2.2). The issue of boundary conditions in Lagrangian models is, on the other hand, less well established and not as well described in the literature.

In the mathematical literature, an Itô-diffusion (of which Eq. (2) is an example) on a domain with a reflecting boundary is known as a Skorokhod-problem, after Ukrainian mathematician Anatoliy V. Skorokhod who wrote two early papers on the topic (Skorokhod, 1961, 1962). In particular, Skorokhod (1961) describes how an SDE with a reflecting boundary can be formally described by adding a term which is zero everywhere except on the boundary, and which causes any trajectory that touches the boundary to be immediately reflected. However, the focus of that paper was to prove the existence and uniqueness of solutions to this modified system, and not on numerical methods.





The classic reference book by Kloeden and Platen (1992) on numerical solution of SDEs does not itself discuss reflecting boundary conditions, but mentions a few references in the section on bibliographical notes (Kloeden and Platen, 1992, pp. 593–594). These are all of a quite technical nature.

In the applied literature on atmospheric transport modelling with random flight models, the issue of boundary conditions has received some attention (see, *e.g.*, Wilson and Flesch (1993) and Rodean (1996, Chapter 11)). Less has been written about boundary conditions in the applied literature on random walk models, which are more commonly used in applied oceanography. For reflecting boundary conditions, a pragmatic and common choice is to simply take any particle that have been randomly displaced to a point outside the boundary, and reflect them to an equal distance inside the boundary (see, e.g., Israelsson et al.

(2006). Some details are discussed in Ross and Sharples (2004), including a numerical artifact that appears when the diffusivity has a non-zero derivative at the boundary. Nordam et al. (2019a) discuss the separate treatment of boundary conditions for advection and diffusion, and conduct a numerical comparison with Neumann and Robin boundary conditions in an Eulerian scheme. In this paper, we will use the same approach as in Nordam et al. (2019a), as summarised below.

We split our numerical scheme (Eq. (A16)) into two parts, and treat the advection and diffusion separately. To implement
no-flux boundary conditions at the sea surface ($z = 0$) for positively buoyant particles, we make the following sequence of steps in the particle model:

- – Random displacement: $z \rightarrow z + K'(z)\Delta t + \sqrt{2K(z)}\Delta W$
- – Reflect about surface: $z \rightarrow -|z|$
- – Rise due to buoyancy: $z \rightarrow z + w\Delta t$
- – Stop particles at surface: $z \rightarrow \min(z, 0)$

For no-flux boundary conditions at the sea floor for negatively buoyant particles, the steps are analogous, but with reflection and stopping at the bottom boundary instead.

To implement a no-diffusive-flux boundary condition, while allowing an advective flux across the boundary at $z = 0$, as discussed in Section 2.4, we instead make the following sequence of steps in the particle model:

- – Random displacement: $z \rightarrow z + K'(z)\Delta t + \sqrt{2K(z)}\Delta W$
- – Reflect about surface: $z \rightarrow -|z|$
- – Rise due to buoyancy: $z \rightarrow z + w\Delta t$
- – Remove particles above surface: $z \rightarrow \text{Surfaced}$

When particles are set to surfaced in the last step, they are removed from the simulation. To get the same boundary conditions
for, *e.g.*, sinking plastics or mineral particles that settle on the sea floor, we again have to reflect about the sea floor in the second step, and removing those particles that have settled out when they cross the sea-floor boundary due to sinking in the last step above. In Nordam et al. (2019a) this approach to implementing boundary conditions in a Lagrangian implementation was found to give identical results to an Eulerian implementation using prescribed fluxes in an FVM.





Depending on the application, particles that have been removed from the simulation may be re-introduced with some prob-
ability, to represent for example breaking waves entraining material from the water surface, or strong currents resuspending
material from the sea floor (see also Sections A2.2 and 4.3).

### 3.2.2 Velocity distribution

In a Lagrangian model, a distribution of terminal rising or sinking velocities can be represented very naturally, simply by
allowing each particle to have its own vertical velocity drawn from the desired distribution. Hence, any distribution can be
represented, and the quality of the representation will depend on the number of particles. The requirement for assigning a
random velocity to each numerical particle is that we can draw samples from the velocity probability distribution. Depending
on the available information, different approaches might be suitable (see, *e.g.*, Press et al. (2007, Chapter 7)). For the three case
studies considered in Section 4, we use three different approaches. A justification for the selection and details are given in the
description for each case.

### 3.2.3 Measures of the numerical error


As described in Section 3.1.3, we will use a direct comparison of particle concentration, as well as the first moment of the
concentration profile, to assess our results. Calculating the first moment (center of mass) of the spatial distribution of particles
is trivial, as this is simply the average position over all particles:

$$M_1(N_p) = \frac{1}{N_p} \sum_{j=1}^{N_p} z_j \qquad (14)$$

where $N_p$ is the number of particles, and $z_j$ is the position of particle $j$. Here we have assumed that all particles represent an
equal mass, but if this is not the case the average is simply weighted by the mass, $m_j$, of each particle.

For the Lagrangian implementation, convergence with the number of particles, $N_p$, is of a stochastic, rather than a deter-
ministic nature. Running a simulation once will give an approximate result for, *e.g.*, the first moment, but with some random
error. Running the simulation again will give another result. To investigate the convergence with the number of particles, we
ran 100 repeated simulations for each value of $N_p$. For each of those 100 simulations, we calculate the first moment, $M_1(N_p)$.
As described in Section 2.6, $M_1(N_p)$ will be a Gaussian random variable whose mean is the true (but unknown) mean of
the distribution, $M_1$, with standard deviation $\sigma(N_p) \sim 1/\sqrt{N_p}$. Hence we consider $\sigma(N_p)$ as a measure of the error when
considering convergence with the number of particles in the Lagrangian implementation.

## 4 Case studies

We present three different cases, and simulate all three with both an Eulerian and a Lagrangian implementation of our model.
Then, we compare the results, and discuss convergence in terms of numerical parameters for both implementations. The cases
are described below, and an overview of some aspects of the cases is presented in Table 1. Table 2 lists the numerical parameters
we have used to investigate convergence.





**Table 1.** A brief summary of the boundary conditions (BC) and reaction terms used in the three different cases.

| Case | System | BC surface | BC sea floor | Reaction terms |
|---|---|---|---|---|
| 1 | Fish eggs | No flux | No flux | No |
| 2 | Microplastics | No flux | No diffusive flux | No |
| 3 | Oil droplets | No diffusive flux | No flux | Droplet entrainment |

**Table 2.** Numerical parameters used for convergence analysis. The same parameters are used in all three cases.

| Scheme | Timesteps, $\Delta t$ [s] | Spatial cells, $N_z$ | Velocity classes, $N_k$ | Particles, $N_p$ | Runs |
|---|---|---|---|---|---|
| Eulerian | 5, 10, 20, 60, 100, 150, 300, 600, 1800 | 500, 750, 1000, 1500, 2000, 3000, 4000, 6000, 8000, 12000, 16000 | 4, 8, 12, 16, 24, 32, 36, 48, 64, 72, 96, 128, 256 | - | 1 |
| Lagrangian | 2, 10, 20, 60, 100, 150, 300, 600, 1800 | - | - | 100, 300, 1000, 3000, 10000, 30000, 100000, 300000, 1000000, 3000000 | 100 |

For case 1, we consider positively buoyant fish eggs, with a distribution of rise speeds. Fish eggs will float to the surface, but

do not leave the water column, and hence we model these with a no-flux boundary condition at the surface (*i.e.*, the particles cannot leave the water column).

For case 2, we consider microplastic particles, with a velocity distribution that includes both rising and sinking particles, representing the diversity of densities associated with different polymer types. In this case, we again use a no-flux boundary at the surface, while the boundary at the seabed is zero-flux in diffusion, but allowing an advective flux to leave the domain. This

represents negatively buoyant microplastic particles that are removed from the water column by settling onto the seabed.

For case 3, we consider oil which is entrained as droplets when a surface slick is broken up by waves. The droplets are all positively buoyant with the same density, but have a size distribution which leads to a distribution of rising speeds. In this case, the boundary condition at the surface is zero-diffusive-flux, while allowing an advective flux representing droplets that merge with the surface slick. Additionally, we include the effect of entrainment of oil from the surface slick by breaking waves in our

model.

A spatially variable diffusivity profile is used in all three cases. To calculate eddy diffusivity as a function of depth, we have used the GOTM turbulence model (Umlauf et al., 2005). The model was forced with a surface stress corresponding to a wind speed of 9 m/s (Gill, 1982, p. 29), and run with a $k$-$\omega$ turbulence closure with a flux condition for turbulent kinetic energy (TKE) at the surface. The TKE flux at the surface accounts for wave breaking (Craig and Banner, 1994).

To support re-gridding and differentiation, an analytical expression has been fitted to the discrete diffusivity output from GOTM. In Fig. 1, the fitted analytical profile is shown together with the output from GOTM. The analytical expression is given



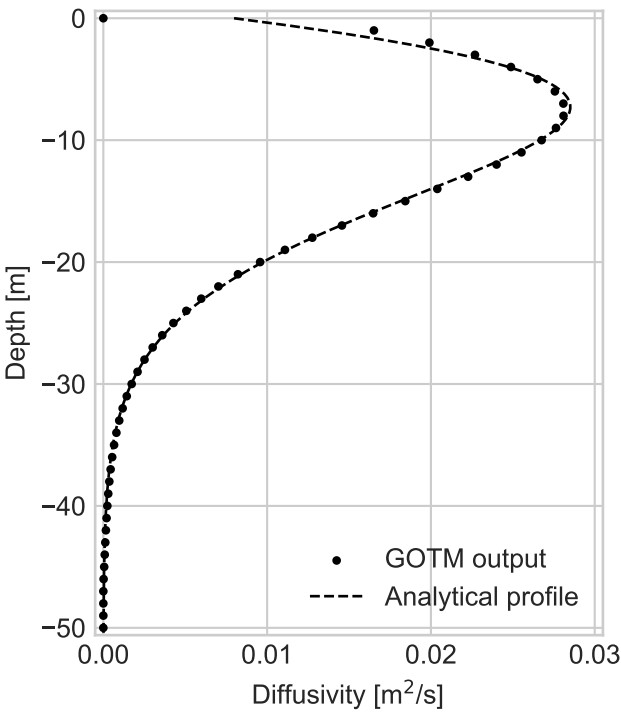

**Figure 1.** Eddy diffusivity output from a GOTM simulation, shown with the fitted analytical diffusivity profile used for the case studies.

by

$$K(z) = \beta (z_0 - z) \cdot e^{-(\gamma (z_0 - z))^{\delta}}, \tag{15}$$

where $\beta = 0.00636 \, \mathrm{m/s}$, $\gamma = 0.088 \, \mathrm{1/m}$, $\delta = 1.54$, $z_0 = 1.3 \, \mathrm{m}$, and $z$ is negative downwards. In the Eulerian implementation,
the diffusivity is given by the value of Eq. (15) evaluated at the cell boundaries, while for the Lagrangian implementation,
Eq. (15) is evaluated at the position of each particle at each timestep.

For each of the three cases, our initial condition will be a submerged particle distribution. The spatial distribution will, in
all cases, be a Gaussian distribution centered at $-20 \, \mathrm{m}$ depth, with a standard deviation of $4 \, \mathrm{m}$. The distribution of rise and
settling speeds will be different for each case, and will be explained in the description of each case below.

In all cases, we will present a direct comparison of the predicted total water-column concentrations at different times, from
the Eulerian and the Lagrangian implementations. Total concentration in this case means integrated over the velocity distri-
bution, showing the total suspended concentration regardless of rising or settling velocity. Additionally, we show a numerical
convergence analysis, separately for the two schemes. For the Eulerian implementation, we consider error as a function of
the number of velocity classes. For the Lagrangian implementation we present error as a function of the number of particles.





As a measure of the error, we consider the first moment (the center of mass) of the spatial concentration distribution (see Sections 3.1.3 and 3.2.3). Additional numerical convergence tests are also presented in Appendix C.

### 4.1 Case 1: Pelagic fish eggs

In Case 1 we consider pelagic fish eggs, with a distribution of rise velocities. These positively buoyant particles will rise towards the surface, but do not form a surface slick. Rather, they rise to the surface in stagnant water, but stay submerged, and

can be mixed back down by the eddy diffusivity (Sundby, 1983; Röhrs et al., 2014; Sundby and Kristiansen, 2015).

Taking an example from Sundby (1983, Table 3), we consider the eggs of the Arcto-Norwegian cod (*Gadhus morhua* L.), which are typically found to have a Gaussian distribution of terminal rise velocities with average $\bar{w} = 0.96\,\mathrm{mm/s}$, and a standard deviation of $\sigma = 0.38\,\mathrm{mm/s}$. We furthermore assume that the distribution is truncated symmetrically about the mean, such that $w \in [\bar{w} \pm 2\sigma]$.

To divide this distribution into $N_k$ velocity classes for use in the Eulerian implementation, we divide the interval $[\bar{w} \pm 2\sigma]$ into $N_k$ sub-intervals with equal spacing $\Delta w$ on a linear scale, and calculate the mass fraction in each class as described in Section 3.1.2. For the Lagrangian implementation, we sample the particle velocities at random from the truncated Gaussian distribution. Hence, a numerical particle may have any rise speed $w \in [\bar{w} \pm 2\sigma]$.

The results of our simulations for fish eggs are shown in Fig. 2. In the left panel, concentration profiles are shown for four

different times (including $t = 0$), with the Lagrangian results shown as continuous coloured lines and the Eulerian results for the same times shown as dashed black lines. The concentration profiles from the Eulerian implementation were produced with timestep $\Delta t_E = 10\,\mathrm{s}$, $N_k = 128$ classes, and $N_z = 8000$ spatial cells, while the Lagrangian results used timestep $\Delta t_L = 10\,\mathrm{s}$ and $N_p = 3 \times 10^6$ particles converted to concentration by bincount on 8000 bins. We observe that the two implementations produce essentially identical predicted concentrations.

In the middle panel of Fig. 2, we show the convergence of the Eulerian results with the number of velocity classes, with the other numerical parameters kept constant at timestep $\Delta t_E = 10\,\mathrm{s}$, and $N_z = 8000$ spatial grid cells. The plot shows the error in the first moment, where the error is calculated relative to a reference solution from a simulation with $N_k = 256$ classes. We observe that the convergence with number of classes appears to be of order 2, with the error going down as $1/N_k^2$.

Finally, in the right-hand panel, we show the convergence of the Lagrangian results as a function of the number of particles,

$N_p$, keeping a fixed timestep of $\Delta t_L = 10\,\mathrm{s}$. The figure shows the standard deviation of the modelled first moment, calculated over an ensemble of 100 simulations for each value of $N_p$. As noted above, the first moment is just the average particle position, and as expected from the Lindeberg-Lévy theorem, the standard deviation of the first moment goes down as $1/\sqrt{N_p}$ (see Section 2.6).

### 4.2 Case 2: Microplastics

In this case, we consider a distribution of microplastic particles. As the only parameter describing the numerical particles in our model is the terminal rise/settling velocity, we have obtained a velocity distribution based on published descriptions of microplastics.





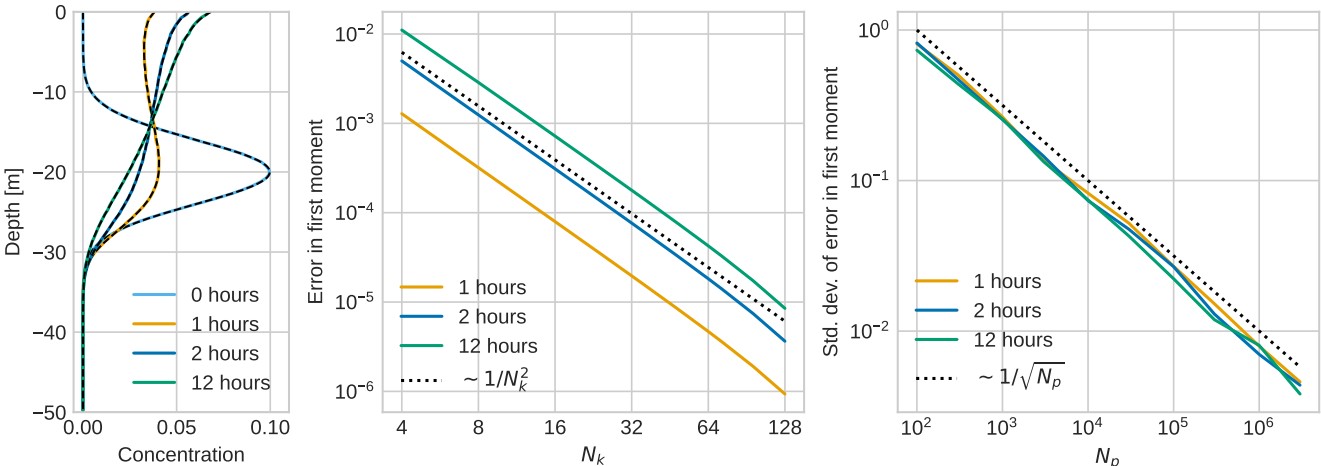

**Figure 2.** Results for Case 1: Fish eggs. The left panel shows concentration profiles, at different times, with Lagrangian results shown as coloured continuous lines and Eulerian results shown as dashed black lines, for the same times. Middle panel: Convergence of the error in the first moment for the Eulerian results, shown as a function of number of velocity classes. Right panel: Convergence of the standard deviation of the first moment for the Lagrangian results, as a function of number of particles.

As our starting point, we consider particles with a distribution of densities from 0.8 kg/L to 1.5 kg/L, a distribution of sizes from 20 $\mu$m to 5 mm, as well as a distribution of shapes described by the Corey Shape Factor. We assume the density of seawater to be 1.025 kg/L. The distributions of density, size and shape are taken from Kooi and Koelmans (2019). The terminal rise/settling velocities of these particles are calculated from empirical relations obtained by Waldschläger and Schüttrumpf (2019), which give terminal velocities as a function of density, size and shape. To obtain a distribution of velocities, we used a Monte Carlo method consisting of generating a large number of random combinations of density, size and shape, drawn from the distributions given in Kooi and Koelmans (2019), and mapping those to velocities via the relationships given by Waldschläger and Schüttrumpf (2019). A detailed description of the steps involved in obtaining the distribution of velocities is given in Appendix B. See also Isachenko (2020) for another example of a similar approach.

For the Eulerian simulations, discretisations of the velocity distribution into different numbers of classes were obtained as histograms with different numbers of bins, of $10^9$ randomly generated terminal velocities. Further details are provided in Appendix B, with the velocity distribution shown in Fig. B1. For the Lagrangian simulations, the terminal velocity of the particles was by directly generating random velocities as described above and in Appendix B.

In this case, particles are allowed to leave the water column via the seafloor at $-50\,\mathrm{m}$, where we enforce zero diffusive flux, while allowing the advective flux to carry particles across the boundary. This represents particles that leave the water column (and thus the simulation) by settling onto the sea floor, where they are no longer able to be resuspended by the eddy diffusivity. Different resuspension mechanisms can of course be included in the model, but we have chosen to to ignore that here.

The results of our simulations for microplastics are shown in Fig. 3. In the left panel, concentration profiles are shown for four different times, with the Lagrangian results shown as continuous coloured lines and the Eulerian results for the same times





shown as dashed black lines. As in Case 1, the concentration profiles from the Eulerian implementation were produced with timestep $\Delta t_E = 10\,\text{s}$, $N_k = 128$ classes, and $N_z = 8000$ spatial cells, while the Lagrangian results used timestep $\Delta t_L = 10\,\text{s}$ and $N_p = 3 \times 10^6$ particles converted to concentration by bincount on 8000 bins. Again, we observe that the two implementa-
tions give very similar predictions, as the curves are visually indistinguishable.

In the middle panel, we show the convergence of the first moment in the Eulerian results as a function of number of classes. The $1/N_k^2$ trend is less clear than in Case 1, and the size of the error appears to be somewhat sensitive to the number of classes, as for example 32 classes give a smaller error than 36 classes for two of the times considered.

In the right panel we show the convergence with number of particles of the standard deviation of the first moment for the
Lagrangian implementation. Again, as expected from the discussion in Section 2.6, this follows a $1/\sqrt{N_p}$ trend.

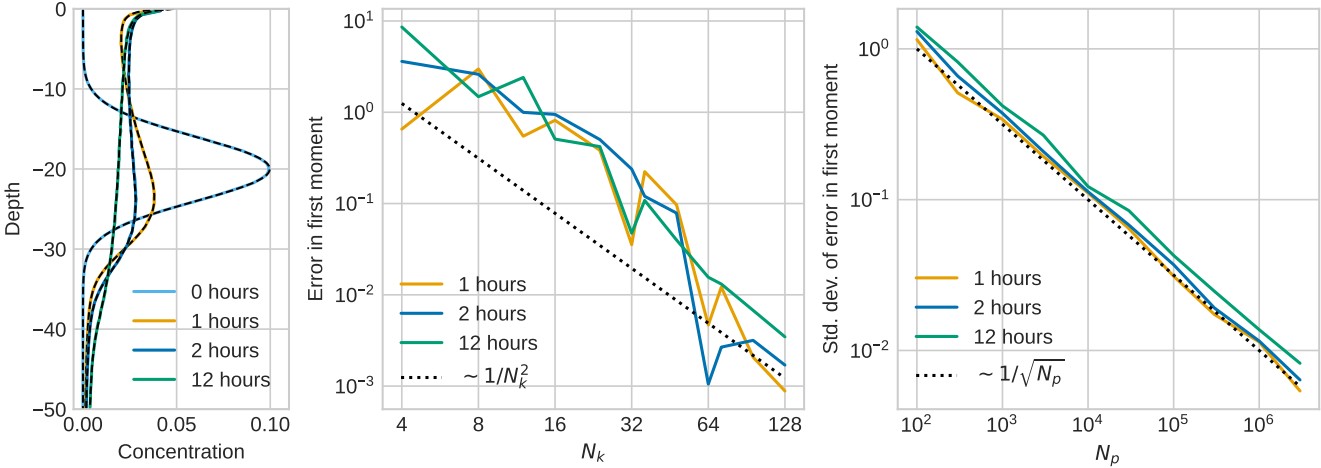

**Figure 3.** Results for Case 2: Microplastics. The left panel shows concentration profiles, at different times, with Lagrangian results shown as coloured continuous lines and Eulerian results shown as dashed black lines, for the same times. Middle panel: Convergence of the error in the first moment for the Eulerian results, shown as a function of the number of velocity classes. Right panel: Convergence of the standard deviation of the error in the first moment for the Lagrangian results, as a function of the number of particles.

In contrast to Case 1, here we have particles with both positive and negative terminal velocities. As the particles are allowed to leave the water column, we also see that the total mass in suspension (*i.e.*, the integral of the concentration profile) decreases with time. Hence, we also present the remaining suspended mass as a function of time. The top panel of Fig. 4 shows the remaining mass fraction suspended in the water column, for both implementations, while the bottom panel shows the difference
between the two implementations for 10 different Lagrangian runs. The oscillatory nature of the difference is due to the randomness of the stochastic Lagrangian particle model. The results were produced with the same numerical parameters as stated in the previous paragraph.



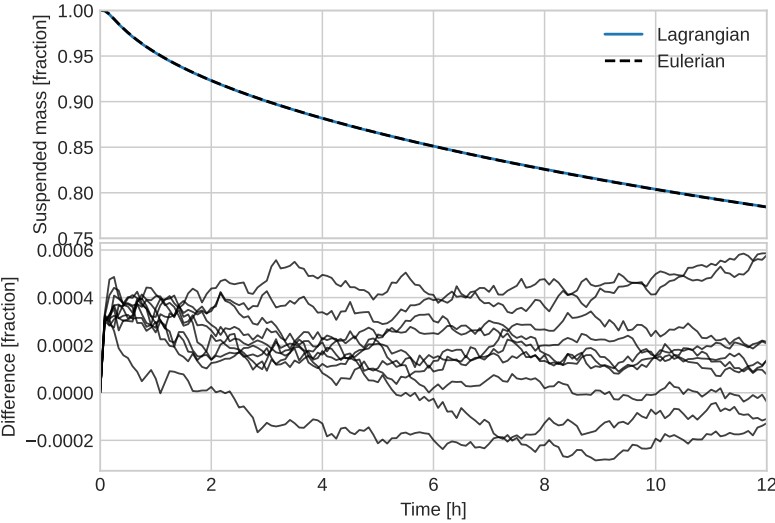

**Figure 4.** Top: Remaining suspended mass as a function of time, shown for the microplastics case for both the Eulerian and the Lagrangian implementation. Bottom panel: The difference between the suspended mass fractions of the two models, shown for 10 different runs of the Lagrangian implementation.

### 4.3 Case 3: Oil droplets with breaking wave entrainment

In this case, we consider a simplified one-dimensional oil spill model, which includes entrainment of surface oil by breaking
waves, rising of oil droplets due to buoyancy, turbulent diffusive mixing, and oil droplets rising to the surface, forming and/or joining a surface slick. As the entrainment mechanism is unique to this case, it is described in some detail below.

To model the entrainment of surface oil by breaking waves, we need the entrainment rate, the entrainment depth, and the size distribution of the entrained droplets. We model entrainment as a first-order decay process (Johansen et al., 2015)

$$\frac{\mathrm{d}Q_s}{\mathrm{d}t} = -\alpha Q_s, \tag{16}$$

where $Q_s$ is the amount of oil at the surface, and $\alpha$ is the entrainment rate coefficient. The coefficient $\alpha = f_{wc}/T_m$ is found from the white-cap coverage fraction, $f_{wc}$ and the mean wave period, $T_m$. For the entrainment depth, we use the classic result by Delvigne and Sweeney (1988), where the entrained oil is evenly distributed across the interval

$$1.15H_s < z < 1.85H_s, \tag{17}$$

with $H_s$ being the significant wave height. The size distribution of the entrained droplets is also taken from Johansen et al.
(2019b), using modified Weber number scaling. The size distribution is log-normal, with a fixed width parameter, and a median size, $d_{50}$, which depends on the wave height and the properties of the oil. For a detailed description of this scheme, as well as the values of the relevant oil properties used here, see Nordam et al. (2019b).





From Eq. (16), we see that the entrainment rate depends on the amount of oil at the surface. Since all the oil in this model is either submerged or at the surface, the amount of oil in the surface slick at time $t$, $Q_s(t)$, can be found as

$$Q_s(t) = Q_{tot} - \int_0^H c(z,t)\,\mathrm{d}z, \tag{18}$$

where $Q_{tot}$ is the total amount of oil present.

Using uniform entrainment throughout the interval $z \in [1.15H_s, 1.85H_s]$ (Delvigne and Sweeney, 1988), we find the source term in the Eulerian advection-diffusion-reaction equation from the entrainment rate and the length of the entrainment interval:

$$S(z, c(z, t_n)) = \begin{cases} \alpha \frac{Q_s(t_n)}{0.7H_s} & \text{if} \qquad 1.15H_s < z < 1.85H_s, \\ 0 & \text{otherwise.} \end{cases} \tag{19}$$

In the Lagrangian implementation, we use that the analytical solution of Eq. (16) is an exponential decay with lifetime $\tau = 1/\alpha$. This means we submerge a fraction of the surface slick at every timestep, with that fraction given by

$$p = 1 - \mathrm{e}^{-\Delta t/\tau}. \tag{20}$$

Thus, entrainment is implemented as a stochastic process, where particles that have surfaced are re-entrained with probability $p$ at every timestep. If entrained, a particle is assigned a depth drawn from a uniform random distribution on the interval $z \in [1.15H_s, 1.85H_s]$, and a random size, drawn from the log-normal size distribution described above.

Surfacing of oil droplets is described as a boundary condition where an advective flux is allowed to pass through the boundary at the surface, while enforcing zero diffusive flux, as described in Sections 2.4, A1.3, and 3.2.1. For additional details on the implementation of surfacing and entrainment of oil in a Lagrangian model, see also Nordam et al. (2019a, b).

The results of our simulations for oil droplets are shown in Fig. 5 In the left panel, concentration profiles are shown for four different times, with the Lagrangian results shown as continuous coloured lines and the Eulerian results for the same times shown as dashed black lines. As in Cases 1 and 2, the concentration profiles from the Eulerian implementation were produced with timestep $\Delta t_E = 10\,\mathrm{s}$, $N_k = 128$ classes, and $N_z = 8000$ spatial cells, while the Lagrangian results used timestep $\Delta t_L = 2\,\mathrm{s}$ and $N_p = 3 \times 10^6$ particles converted to concentration by bincount on 8000 bins.

In the middle panel, we show the convergence of the Eulerian results as a function of number of classes. As in Case 1, the error scales with $1/N_k^2$, though here the absolute value of the error is about two orders of magnitude larger. The right panel shows the convergence of the Lagrangian results with the number of particles, which, as before, follows the $1/\sqrt{N_p}$ trend closely.

For this case, the total amount of submerged oil changes over time as oil droplets surface and are re-entrained. Over time, the submerged size distribution shifts towards smaller droplets, as these take longer to reach the surface, causing the center of mass to shift downwards over time. As before, the two schemes give very similar results. As for Case 2, we also here present the remaining suspended mass as a function of time. The top panel of Fig. 6 shows the remaining mass fraction suspended in the water column, for both implementations, while the bottom panel shows the difference between the implementations, for





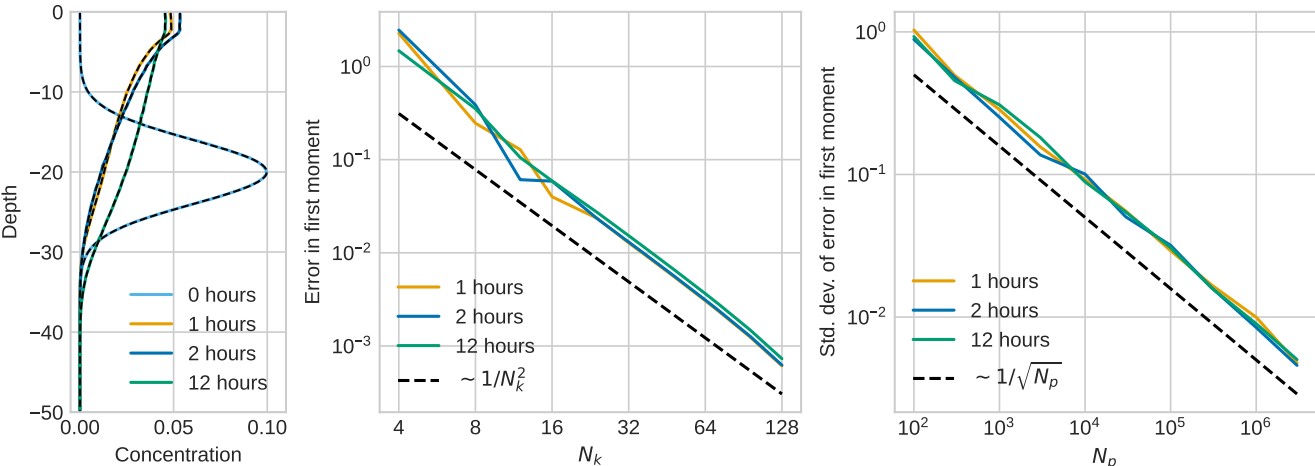

**Figure 5.** Results for Case 3: Oil droplets with entrainment. The left panel shows concentration profiles, at different times, with Lagrangian results shown as coloured continuous lines and Eulerian results shown as dashed black lines, for the same times. Middle panel: Convergence of the error in the first moment for the Eulerian results, shown as a function of the number of velocity classes. Right panel: Convergence of the standard deviation of the first moment for the Lagrangian results, as a function of the number of particles.

10 different runs of the Lagrangian implementation. The oscillatory nature of the difference is due to the randomness of the stochastic Lagrangian particle model. The results were produced with the same numerical parameters as stated in the previous
paragraph.

## 5 Discussion

In this paper, we have conducted a comparison of an Eulerian and a Lagrangian implementation of a water-column model for particles with different distributions of rising and sinking speeds. To highlight different choices of boundary conditions and a reaction term, we have chosen to use fish eggs, microplastics, and oil droplets as our example cases. The model can also easily
be applied to other cases, such as mineral particles, nanoparticles, algae, and chemicals. More complex reaction terms, such as agglomeration, can also be added to the same modelling framework.

### 5.1 Boundary conditions

Boundary conditions are always an essential part of the problem for any model based on PDEs, and indeed the boundary conditions must be specified before the problem can be said to be "well posed" (Gustafsson, 2008, Chapter 2). The implementation
of different boundary conditions in Eulerian models is amply addressed in the applied numerical literature (see, *e.g.*, Hundsdorfer and Verwer (2003); Versteeg and Malalasekera (2007)). Hence, any Eulerian models for environmental transport problems may draw on a wide body of standard reference works describing the implementation of different boundary conditions.

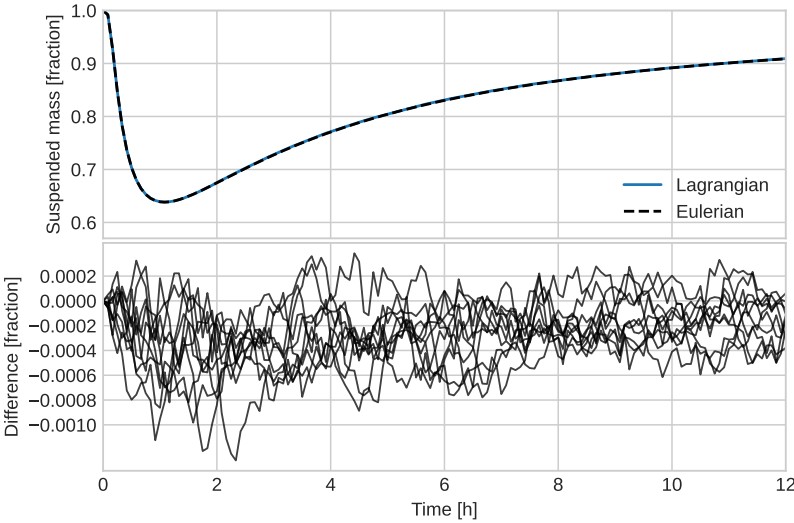

**Figure 6.** Top: Remaining suspended mass as a function of time, shown for the oil case for both the Eulerian and the Lagrangian implementation. Bottom panel: The difference between the suspended mass fractions of the two models.

Transport modelling with stochastic particle methods, on the other hand, is perhaps more of a niche endeavour, and there exists less general applied literature on the topic of numerical methods for SDEs, compared to PDEs. The standard reference
on numerical solution of SDEs, Kloeden and Platen (1992)[1], does not directly address the question of boundary conditions, and in the applied literature the topic is rarely addressed in any detail.

Here, we have provided a detailed description of our implementation of two different types of boundary conditions in the Lagrangian scheme. While our implementation may not have a rigorous foundation in the theory of SDEs, we do present a comparison between our Lagrangian and Eulerian implementations, where the Eulerian method uses a standard approach for
FVMs. In Cases 2 and 3 (microplastics and oil droplets), we show that the two implementations give very similar prediction of the amount of mass which remains suspended in the water column as a function of time, which of course indicates that the net effect of the boundary conditions are the same in the two implementations. For additional discussion of this topic, see also Nordam et al. (2019a).

## 5.2 Convergence of numerical results

We have paid special attention to the representation of particles with a distribution of terminal rising and settling velocities. In a Lagrangian particle model, a distribution of terminal velocities is easily and naturally represented, since each numerical particle can have its own velocity. In an Eulerian model, on the other hand, it is necessary to introduce discrete classes with different velocities to represent the true distribution.

---

[1]More than 11 000 citations, according to Google Scholar (2023).





As a measure of the error in the Lagrangian implementation, we have used the standard deviation of the first moment of the position distribution, $M_1$ (*i.e.*, the center of mass). As is usual for a Monte Carlo method, the standard deviation of $M_1(N_p)$ goes down proportionally to $1/\sqrt{N_p}$, where $N_p$ is the number of particles. If the particles can be considered as independent samples, this behaviour follows from the Lindeberg-Lévy CLT, as discussed in section 2.6. We note, however, that the usual $1/\sqrt{N_p}$ scaling seems to be followed very closely also in Case 3, where the particles cannot strictly be said to represent independent samples due to the inclusion of a distribution-dependent reaction term.

For the Eulerian implementation, we have considered the error in the modelled first moment, $M_1(N_k)$, as a function of the number of classes, $N_k$, used to discretise the velocity distribution representing the particles. We found that the error scales as $1/N_k^2$, where the velocity of each class is represented by the midpoint of the interval spanned by the class. To the best of our knowledge, this result has not previously been published in this context. Models where a distribution of velocities (e.g., due to a distribution of sizes) is represented by a finite number of discrete classes are often used in both marine and atmospheric transport modelling (see, *e.g.*, Tegen and Lacis (1996); Zender et al. (2003); Wichmann et al. (2019); Cui et al. (2020)). Hence, additional understanding of how the number of velocity classes affects accuracy of predictions should be of some interest within these fields.

We note, however, that the result showing $1/N_k^2$ scaling of the error is perhaps not so surprising. Consider as an example the mean of a velocity distribution, $p(v)$, which is given by

$$\bar{v} = \int_{-\infty}^{\infty} v\,p(v)\,\mathrm{d}v. \tag{21}$$

If the integral is evaluated numerically there will be a numerical error, which will depend on the chosen approach. Our approach has been to divide the underlying velocity distribution into equal-sized bins, and let each bin be represented by its midpoint (on either linear or logarithmic scales, depending on the case). This is equivalent to the midpoint quadrature method, which is known to have an error proportional to $\Delta x^2$, where $\Delta x$ is the bin size (Wendroff, 1969, p. 23). This, in turn, is of course proportional to $1/N_k^2$ where $N_k$ is the number of bins.

## 5.3 Relative merits of Eulerian and Lagrangian methods

Numerous studies have discussed different aspects and applications of Lagrangian and Eulerian methods, and compared the two approaches (see, *e.g.*, Fay et al. (1995); Benson et al. (2017); Nordam et al. (2019a); Nepstad et al. (2022)). Here, we consider the question of which implementation should be preferred when modelling particles with a distribution of rising or settling speeds.

Considering first Case 1, we were modelling positively buoyant fish eggs with a Gaussian distribution of terminal rise velocities. For the Eulerian implementation, the numerical error in the first moment due to a finite number of velocity bins starts out between $1 \times 10^{-2}$ m to $1 \times 10^{-3}$ m, for $N_k = 4$ size classes, and reduces very consistently as $1/N_k^2$ from there (see Fig. 2). A numerical error of less than $1\,\mathrm{cm}$ in the position of the center of mass is clearly far smaller than the error due to





other uncertainties and model approximations. Hence, with such a well-behaved velocity distribution in this particular case, an Eulerian model with a small number of velocity classes will give more than adequate numerical accuracy.

For the Lagrangian implementation, on the other hand, we need to use $1 \times 10^6$ particles to reach an error of about $1 \times 10^{-2}$ m, a result of the slower $1/\sqrt{N_p}$ convergence. For modelling fish eggs in a water-column model, an Eulerian approach seems to be the most efficient choice in terms of balance between error and computational effort. This is not necessarily the case in a

full three-dimensional model due to the computational effort involved in tracking the concentration fields for the individual velocity classes across the entire computational domain.

In Case 2 with the microplastics, we have a more complicated velocity distribution. The positive and negative parts of the distribution are shown separately on log scales in Fig. B1, where we see that the positively buoyant part of the distribution is bimodal. Due to the very wide range of velocities present, we chose to use log-spaced velocity bins (separate for positive and

negative velocities) in the Eulerian implementation. Nevertheless, we see from Fig. 3 that the error in the first moment shows some oscillation with changing number of classes. We also note that the absolute value of the error is far higher than in Case 1, and we need around 64 velocity classes to reach an error of $1 \times 10^{-2}$ m in the modelled first moment, $M_1(N_k)$. Compare this to Case 1, where the same accuracy was achieved with only 4 classes.

For the Lagrangian implementation, the results of Case 2 look far more similar to those of Case 1, and we find that $1 \times 10^6$

to $3 \times 10^6$ particles will yield a sampling error of $1 \times 10^{-2}$ m. In the one-dimensional case, the Eulerian implementation is still more efficient than the Lagrangian implementation, but in three dimensions, this is probably no longer the case. The results will, of course, depend on parameters such as timestep, grid resolution and the choice of numerical solver. However, it is clear that the computational effort of solving the advection-diffusion equation for 64 individual velocity classes is considerable.

In Case 3 with the oil droplets, we consider a simplified one-dimensional oil spill model. Lagrangian approaches have a long

history in oil spill modelling, both for horizontal transport (Tayfun and Wang, 1973) and vertical transport (Elliott et al., 1986). A critical issue in oil spill modelling is that the droplet size distribution is both broad and changing time. This is easily modelled with a Lagrangian approach, while an Eulerian approach might require either some form of dynamic size class scheme, or a very large number of size classes. From the middle panel of Fig. 5, we see that the error in the first moment starts at a little above 1 m for $N_k = 4$ classes, and goes down fairly consistently as $1/N_k^2$. As such, the scaling is very similar to Case 1,

while the prefactor is again 2-3 orders of magnitude larger and an error of $1 \times 10^{-2}$ m is achieved with $N_k = 32$ classes. The Lagrangian approach, on the other hand, behaves quite similarly to Cases 1 and 2, and reaches an error of $1 \times 10^{-2}$ m with $1 \times 10^6$ particles.

It is important to point out that the errors discussed above are of different natures for the Eulerian and the Lagrangian implementations. In the Eulerian approach, solving with a finite number of speed classes, $N_k$, leads to a systematic error that

goes to zero as $N_k \to \infty$. In contrast, a finite number of particles, $N_p$, leads to a stochastic sampling error in the Lagrangian approach. If the particles are independent, the sampling error is *not* systematic, and when $N_p \to \infty$, the standard deviation of the sample error goes to zero as $1/\sqrt{N_p}$. See also further discussion on this point in Section 2.6 and Appendix C.





# 6 Conclusions

In this paper, we have implemented and compared two different versions of a water-column model for particles that undergo
diffusion, and which rise or sink with a wide distribution of terminal velocities. Our Eulerian implementation used discrete
velocity classes to represent the velocity distribution, while the Lagrangian implementation allowed each particle to have its
own velocity.

We have studied the rate of convergence of the two different implementations, considering the center of mass (*i.e.*, the first
moment of the concentration profile) as a measure of the error. Our main interest has been to show that we can implement dif-
ferent boundary conditions in an equivalent manner in the two schemes, and to demonstrate how the numerical error is reduced
with an increasing number of velocity classes and particles for the Eulerian and Lagrangian implementations, respectively.
Convergence results for varying timestep and (in the Eulerian case) spatial resolution are also shown in Appendix C.

Three different example cases were considered: Positively buoyant fish eggs with a Gaussian velocity distribution, mi-
croplastics with a distribution of positive and negative velocities obtained by a Monte Carlo approach (see Appendix B), and
positively buoyant oil droplets with a time-varying velocity distribution. In all three cases, we find that the stochastic sampling
error in the Lagrangian method scales as $1/\sqrt{N_p}$ (where $N_p$ is number of particles) with about the same prefactor for all cases.
Interestingly, this also appears to hold in the case of oil droplets, where the particles cannot strictly be said to be independent
samples.

For the Eulerian implementation, we find that the error appears to scale as $1/N_k^2$ (where $N_k$ is the number of velocity classes),
though the prefactor varies by several orders of magnitude between the cases. We also see that in the case of microplastics, the
exact choice of the velocity classes has a significant impact on the error, possibly due to the multi-modal nature of the velocity
distribution.

Owing to current shear and density gradients, vertical behaviour is very important for correctly modelling horizontal trans-
port in the ocean. While it is hard to draw strict conclusions about three-dimensional simulations from one-dimensional ex-
amples, it should still be clear that the number of classes needed to get accurate results in an Eulerian simulation will depend
strongly on the velocity distribution of the relevant particles. We observed that far fewer classes were needed for the fish eggs
case, with a Gaussian velocity distribution, than for the microplastics case, which used a wider, multi-modal distribution. Sim-
ilarly, the time-dependent size distribution (and thus velocity distribution) seen in the oil case needed more classes to give
accurate results than the fish egg case. Hence, our results would suggest that Lagrangian particle models have a particular
advantage when wider, less "normal" velocity distributions are considered, or when the size distribution changes with time.

*Code availability.* The code used to run the simulations and create the plots shown in this paper is available under a GPL-3.0 license, and
can be found at https://github.com/nordam/Eulerian-and-Lagrangian-methods or at https://doi.org/10.5281/zenodo.7851010

.





**Figure A1.** Illustration of the discretisation of the $z$ axis in the Eulerian implementation. Concentrations are calculated at cell centers (marked with round dots), with integer indices, while fluxes are calculated through cell faces (marked with vertical ticks), at half-integer indices.

**Appendix A: Additional implementation details**

**A1 Eulerian implementation**

**A1.1 Discretisation**

The advection-diffusion-reaction (ADR) equation describes how the concentration, $C_k$, of a substance affected by diffusive, advective and reactive processes develops in time. In this study, we consider the one-dimensional version of the ADR equation, and consider vertical motion only. The diffusivity is assumed to be a function of position. Similarly, the advection velocity,

which here represents particles rising or sinking due to buoyancy, may also be spatially dependent such that particles stop at the surface and/or bottom boundaries. Finally, the reaction term is assumed to depend on both concentration and position. Consequently, Eq. (1) with all dependencies explicitly shown becomes:

$$\frac{\partial C_k(z,t)}{\partial t} = \frac{\partial}{\partial z}\left(K(z)\frac{\partial C_k(z,t)}{\partial z}\right) - \frac{\partial}{\partial z}\big(w_k(z)C_k(z,t)\big) + S\big(z, C_1(z,t),\ldots,C_{N_k}(z,t)\big), \tag{A1}$$

where $C_k(z,t)$ is the concentration of material of class $k$, $K(z)$ is the spatially dependent diffusivity, $w_k(z)$ is the spatially de-
pendent velocity of material in class $k$, $S\big(z, C_1(z,t),\ldots,C_{N_k}(z,t)\big)$ is the source term (which may depend on the concentration of all classes) and $N_k$ is the number of classes.

In Finite Volume Methods (FVMs), the PDE is converted to integral form by integration over control volumes, or cells. As illustrated in Fig. A1, cell centers are given integer indices, and cell faces half-integer indices, such that cell $j$ extends from $z_{j-1/2}$ to $z_{j+1/2}$, with cell width equal to $\Delta z_j$. Considering a spatially distcretised version of Eq. (A1), we use the divergence

theorem to write the volume integrals of the terms on the right-hand side as surface integrals over the cell faces. By applying the Leibniz integral rule to change the order of the spatial integral and the differentiation with respect to time on the left-hand side, the average concentration of class $k$ in cell $j$, $\overline{C}_{k,j}$, may be defined. Similarly the average reaction term for class $k$ in cell $j$, $\overline{S}_{k,j}$, is found by integration.

The integral form of Eq. (A1), for the average concentration of material of class $k$ in cell $j$ thus reads

$$\Delta z_j \frac{\mathrm{d}\overline{C}_{k,j}}{\mathrm{d}t} = \left(K\frac{\partial C_k}{\partial z}\right)_{j+\frac{1}{2}} - \left(K\frac{\partial C_k}{\partial z}\right)_{j-\frac{1}{2}} - \big(w_k C_k\big)_{j+\frac{1}{2}} + \big(w_k C_k\big)_{j-\frac{1}{2}} + \Delta z_j \overline{S}_{k,j}, \tag{A2}$$

where the subscripts on the brackets indicate where the variables are evaluated, *e.g.*, $(w_k C_k)_{j+1/2} = w_k(z_{j+1/2})C_k(z_{j+1/2})$. This is noted to be an exact equation for the rate of change of the cell-averaged concentrations, $\overline{C}_{k,j}$, given in terms of the



advective and diffusive fluxes through the cell faces, and the cell-average source term. The fact that the concentration within a cell is explicitly expressed as the sum of the fluxes into and out of the cell gives the FVM its mass-conserving properties (in the case when $S = 0$).

Next, we make an approximation by assuming the concentration to be constant within each control volume, such that the cell-averages, $\overline{C}_{k,j}$ and $\overline{S}_{k,j}$ may be approximated by the values located at the cell centres. The concentration and source term at $z_j$ are denoted $C_{k,j}$ and $S_{k,j}$, respectively. Choosing a constant grid spacing $\Delta z$, the $z$-axis is discretised into $N_z = H/\Delta z$ equally sized cells, where $H$ is the length of the domain. For another example of a similar derivation, see Versteeg and Malalasekera (2007, pp. 243-246).

### A1.2    Numerical approximation of fluxes

In Eq. (A2), the first two terms on the right represent the diffusive fluxes through the faces of cell $j$, and the next two terms represent the advective fluxes. The diffusive fluxes at the cell faces are approximated by a second-order central difference of the two adjacent cells (see, *e.g.*, Versteeg and Malalasekera (2007, p. 117)). That is, on the form

$$\left(K\frac{\partial C_k}{\partial z}\right)_{j+\frac{1}{2}} \approx K_{j+\frac{1}{2}}\frac{C_{k,j+1} - C_{k,j}}{z_{j+1} - z_j} = K_{j+\frac{1}{2}}\frac{C_{k,j+1} - C_{k,j}}{\Delta z}, \tag{A3}$$

where the diffusivity $K_{j+\frac{1}{2}}$ is determined on the cell faces either explicitly by an analytic expression, if available, or by interpolation in the case of a discrete diffusivity.

For the advective fluxes, however, linear numerical schemes of second-order accuracy and higher are known to yield numerical oscillations for advection-dominated problems (*i.e.*, $\mathrm{Pe} = \frac{|v|\Delta z}{K} \gg 1$), for non-smooth solutions (Hundsdorfer and Verwer, 2003, pp. 66-67, 118-119). To be able to handle advection-dominated cases, we implemented a more flexible second-order numerical scheme for advection. A flux-limiter approach was used, where we approximate the advective fluxes with the first-order upwind scheme with a second-order correction which depends on a limiter function (Hundsdorfer and Verwer (2003, pp. 216–217); Versteeg and Malalasekera (2007, pp. 165-171)). The advective fluxes are thus approximated as

$$(w_k c_k)_{j+\frac{1}{2}} \approx w_{k,j+\frac{1}{2}}^+\left(C_{k,j} + \frac{1}{2}\psi(\rho_{j+\frac{1}{2}}^+)(C_{k,j+1} - C_{k,j})\right) + w_{k,j+\frac{1}{2}}^-\left(C_{k,j+1} - \frac{1}{2}\psi(\rho_{j+\frac{1}{2}}^-)(C_{k,j+1} - C_{k,j})\right). \tag{A4}$$

Here, positive and negative velocities are handled separately by letting $w_{k,j+1/2}^+ = \max(0, w_k(z_{j+1/2}))$ and $w_{k,j+1/2}^- = \min(0, w_k(z_{j+1/2}))$. The limiter function, $\psi(\rho^\pm)$, determines the correction, with $\rho^+$ and $\rho^-$ given by

$$\rho_{j+\frac{1}{2}}^+ = \frac{C_{k,j} - C_{k,j-1}}{C_{k,j+1} - C_{k,j}} \tag{A5a}$$

$$\rho_{j+\frac{1}{2}}^- = \frac{C_{k,j+2} - C_{k,j+1}}{C_{k,j+1} - C_{k,j}}. \tag{A5b}$$

The flux limiting was done with the UMIST limiter function (Lien and Leschziner, 1994; Versteeg and Malalasekera, 2007, pp. 170–178), given by

$$\psi(\rho) = \max\left[0, \min\left(2\rho, \frac{1+3\rho}{4}, \frac{3+\rho}{4}, 2\right)\right]. \tag{A6}$$





We see that $\rho_{j+1/2}^{\pm}$ is the ratio between the concentration gradients at the upstream or downstream sides, and the gradient across the cell face at $z_{j+1/2}$ (Versteeg and Malalasekera (2007, pp. 167, 171-172); Hundsdorfer and Verwer (2003, p. 216)). If $\rho$ is close to 1, which will be the case for reasonably smooth concentration profiles, then $\psi(\rho)$ is also close to 1, in which
case Eq. (A4) is approximately equal to a regular second-order central finite difference. On the other hand, if there are large differences in concentration gradients between neighbouring pairs of cells, then $\psi(\rho)$ will be close to either 0 or 2, in which case Eq. (A4) will be approximately equal to a first-order upwind or downwind scheme, to avoid numerical oscillations. It also reduces to upwind near the boundaries. This approach was found to result in second-order accuracy in space for the cases considered here (see Appendix C for examples of grid-refinement tests).

### 660 A1.3 Boundary conditions

As described in Section 2.4, we consider two different types of boundary conditions, zero-flux (Eq. (6)) and zero-diffusive-flux (Eq. (7)). In our Eulerian implementation, which uses a finite-volume discretisation scheme, it is straightforward to use different boundary conditions by simply setting the required fluxes to zero.

First, we consider the cell adjacent to the surface boundary. Note that the uppermost cell is indexed $j = 0$, and that the cell
center is at $z_0 = \Delta z/2$, while the surface boundary is found at $z_{-1/2} = 0$ (see also Fig. A1). The insertion of the zero-diffusive-flux surface boundary condition of Eq. (7) into Eq. (A2) yields

$$\Delta z \frac{\mathrm{d}C_{k,0}}{\mathrm{d}t} = \left(K \frac{\partial C_k}{\partial z}\right)_{\frac{1}{2}} - \left(w_k C_k\right)_{\frac{1}{2}} + \left(w_k C_k\right)_{-\frac{1}{2}} + \Delta z S_{k,0}. \tag{A7}$$

We note that setting $\left(K \frac{\partial C_k}{\partial z}\right)_{-\frac{1}{2}} = 0$ implicitly means that we are assuming zero concentration gradient across the surface, since $K(z) > 0$ everywhere.

To implement a zero-flux boundary condition, we force both the diffusive and advective fluxes to be zero. This we can do by setting the diffusive flux to zero as above, and additionally setting $w_k(z_{-1/2}) = 0$. We note that in our model, the advection velocity represents the rise or settling velocity of the suspended matter, and hence this is equivalent to introducing a spatially variable velocity, saying that a positively buoyant particle has a rise velocity of zero if it is at the surface:

$$w_k(z) = \begin{cases} 0 & \text{if} & z = 0, \\ w_k & \text{otherwise.} \end{cases} \tag{A8}$$

The same reasoning applies to the boundary conditions at the sea floor.

### A1.4 Reaction terms

Reaction terms can be used for different purposes, including removal or addition of mass, or redistribution of mass from one class to another. In this study, we will only consider reaction terms that add mass. We observe from Eqs. (A1) and (A2) that the reaction term, $S$, has units equal to the time-derivative of the concentration, and that it enters our numerical scheme in a
straightforward manner as a rate of change in the average concentration inside a cell. As only one of the three cases we consider make use of a reaction term, the details are presented in Section 4.3.





### A1.5 Numerically solving the discretised equation

The scheme was discretised in time with a fixed timestep, such that

$$t_i = t_0 + i\Delta t, \tag{A9}$$

and integrated by the Crank-Nicolson method (Gustafsson, 2008, p. 39), given by

$$\frac{C_{k,j}^{i+1} - C_{k,j}^{i}}{\Delta t} = \frac{1}{2}F_{k,j}^{i} + \frac{1}{2}F_{k,j}^{i+1}, \tag{A10}$$

where $C_{k,j}^{i}$ is the concentration of class $k$ in cell $j$ at time $t_i$, and $F_{k,j}^{i}$ is the right-hand side of Eq. (A2) evaluated at time $t_i$. The Crank-Nicolson scheme was chosen for its second-order accuracy in time and favourable stability (Versteeg and Malalasekera, 2007, pp. 247-248).

With the chosen discretisation schemes in space and time, our numerical solver can be rewritten into a matrix equation on the form

$$\mathbf{L}\mathbf{C}^{i+1} = \mathbf{R}\mathbf{C}^{i} + \frac{\mathbf{S}^{i} + \mathbf{S}^{i+1}}{2}, \tag{A11}$$

where $\mathbf{L}$ and $\mathbf{R}$ are matrices to be described below. The vectors $\mathbf{C}^{i}$ and $\mathbf{S}^{i}$ contain respectively the concentration in each cell for each class, and the source term for each cell and each class, at time $t_i$ (see schematic illustration in Fig. A2).

The matrices $\mathbf{L}$ and $\mathbf{R}$ can each be written as sums of two terms:

$$\mathbf{L} = \mathbf{L}_{AD} + \mathbf{L}_{FL}, \quad \mathbf{R} = \mathbf{R}_{AD} + \mathbf{R}_{FL}. \tag{A12}$$

Here $\mathbf{L}_{AD}$ and $\mathbf{R}_{AD}$ contain the coefficients that implement the finite difference approximations of the spatial derivatives of the concentration, specifically the central finite difference for diffusion, and upwind for advection. $\mathbf{L}_{FL}$ and $\mathbf{R}_{FL}$ contain the coefficients for the flux-limiter correction to the advection scheme.

In all the cases we consider below, the matrices describing the advection and diffusion terms remain constant in time. The flux limiter correction and the reaction terms, on the other hand, are themselves functions of the concentration, $\mathbf{C}$. Since this leads to a non-linear system of equations, the equation system given by Eq. (A11) cannot be solved directly as in the linear case, and thus we have adopted an iterative scheme.

Writing out Eq. (A11) with dependence on concentration shown explicitly, we get

$$\left(\mathbf{L}_{AD} + \mathbf{L}_{FL}(\mathbf{C}^{i+1})\right)\mathbf{C}^{i+1} = \left(\mathbf{R}_{AD} + \mathbf{R}_{FL}(\mathbf{C}^{i})\right)\mathbf{C}^{i} + \frac{\mathbf{S}(\mathbf{C}^{i}) + \mathbf{S}(\mathbf{C}^{i+1})}{2}. \tag{A13}$$

This equation must be solved at every timestep to find the concentration, $\mathbf{C}^{i+1}$, at time $t_{i+1}$, from the current concentration, $\mathbf{C}^{i}$, at time $t_i$, taking the reaction terms $\mathbf{S}^{i}$ and $\mathbf{S}^{i+1}$ into account. Since $\mathbf{C}^{i+1}$ is of course not yet known at time $t_i$, we start by using the current concentration as an initial guess at the next concentration, $\tilde{\mathbf{C}}^{i+1} = \mathbf{C}^{i}$, and then we solve the system

$$\left(\mathbf{L}_{AD} + \mathbf{L}_{FL}(\tilde{\mathbf{C}}^{i+1})\right)\tilde{\tilde{\mathbf{C}}}^{i+1} = \left(\mathbf{R}_{AD} + \mathbf{R}_{FL}(\mathbf{C}^{i})\right)\mathbf{C}^{i} + \frac{\mathbf{S}(\mathbf{C}^{i}) + \mathbf{S}(\tilde{\mathbf{C}}^{i+1})}{2}. \tag{A14}$$





We then refine our guess by letting $\tilde{\mathbf{C}}^{i+1} \to \tilde{\tilde{\mathbf{C}}}^{i+1}$, solve again, and repeat. At each iteration we calculate a measure of the error,

$$\text{err} = \max_{k \in [1, N_k]} \left\| \tilde{\mathbf{C}}_k^{i+1} - \tilde{\tilde{\mathbf{C}}}_k^{i+1} \right\|_2, \tag{A15}$$

and we terminate the iterative procedure when $\text{err} < \eta \cdot \text{err}_0$, where $\text{err}_0$ is the error calculated from the first guess, for some tolerance $\eta$.

For the cases we consider, $\mathbf{L}$ and $\mathbf{R}$ are tri-diagonal matrices, which means that Eq. (A14) can be solved efficiently using the tri-diagonal matrix algorithm (TDMA, see for example Press et al. (2007, pp. 56–57)).

### A1.6 Solving for multiple classes

Equation (A1) describes the advection and diffusion of a concentration field $C_k(z,t)$, where the advection velocity, $w_k$, represents the rising or settling speed of particles of class $k$. In this paper, we wish to consider a distribution of particles with different rising or settling speeds, represented by discrete classes.

If there are no reaction terms, there is no interaction between the different classes of particles, and hence the advection-diffusion equation can be solved independently for each class as described in the previous section. Nevertheless, for reasons of flexibility, it may be convenient to implement an approach where the advection-diffusion equation is solved simultaneously for all classes, since this allows interacting reaction terms to be added.

When there are reaction terms that allow the classes to interact, *i.e.*, if the reaction term $S_k$ of class $k$ depends on the concentration of any other class, then we have to solve the advection-diffusion-reaction equation simultaneously for all classes.

Our chosen approach is first to set up the equation system for each individual class. The concentration of class $k$ in the different spatial cells is then represented by a vector $\mathbf{C}_k$, and the matrices in Eq. (A11) for class $k$ we call $\mathbf{L}_k$ and $\mathbf{R}_k$. We then create the full matrices $\mathbf{L}$ and $\mathbf{R}$ as block-diagonal matrices with the blocks being made up of the matrices $\mathbf{L}_1, \ldots, \mathbf{L}_{N_k}$, and $\mathbf{R}_1, \ldots, \mathbf{R}_{N_k}$. We next combine the concentrations for all the classes into an overall vector $\mathbf{C} = [\mathbf{C}_1, \ldots, \mathbf{C}_{N_k}]$, where $N_k$ is the total number of classes. This leads to a linear system of equations, schematically illustrated in Fig. A2, which can be solved to find the concentrations in all cells for all classes.

As a practical matter, we note that since the matrices $\mathbf{L}$ and $\mathbf{R}$ are tridiagonal, using a sparse array class in the implementation means that the full matrices do not require any additional memory compared to storing the matrices $\mathbf{L}_1, \ldots, \mathbf{L}_{N_k}$, and $\mathbf{R}_1, \ldots, \mathbf{R}_{N_k}$ separately.

### A2 Lagrangian implementation

#### A2.1 Numerical solution of the SDE

We solve our stochastic differential equation with the Euler-Maruyama method (Kloeden and Platen, 1992, p. 305). With this method, the discretised version of Eq. (2) becomes

$$z_{i+1} = z_i + \left(w + K'(z_i)\right)\Delta t + \sqrt{2K(z)}\,\Delta W_i. \tag{A16}$$



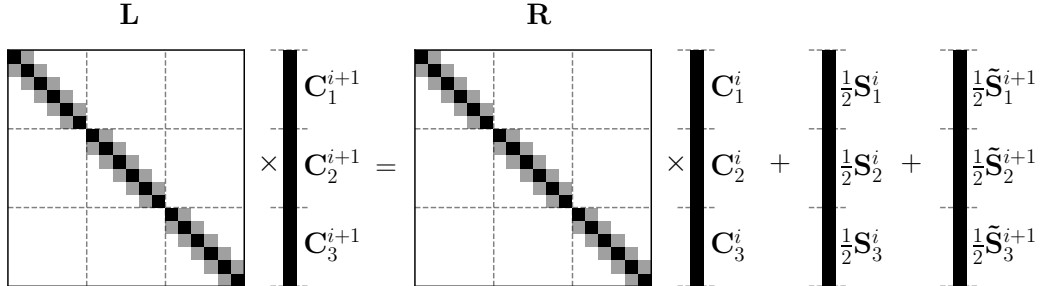

**Figure A2.** Schematic illustration of Eq. (A11), for a case with $N_z = 6$ spatial cells and $N_k = 3$ classes. The matrices **L** and **R** have a block diagonal structure where each block is itself tridiagonal. The vectors $\mathbf{C}^i$ and $\mathbf{C}^{i+1}$ are split up into sub-vectors $\mathbf{C}_1, \ldots, \mathbf{C}_{N_k}$, where the elements of $\mathbf{C}_k$ are the concentrations of class $k$ in the different spatial cells. Similarly, the reaction term vectors $\mathbf{S}^i$ and $\tilde{\mathbf{S}}^{i+1}$ are split into sub-vectors in the same way.

Here, $\Delta W_i$ are independent Gaussian distributed random numbers with zero mean, $\langle \Delta W_i \rangle = 0$, and variance $\langle \Delta W_i^2 \rangle = \Delta t$.

The Euler-Maruyama method has order of convergence 1 in the weak sense, which means that for sufficiently small $\Delta t$, we have that

$$|\langle f(z_n) \rangle - \langle f(z(t_n)) \rangle| \leq \gamma \Delta t, \tag{A17}$$

where $z(t_n)$ is the true (but usually unknown) solution at time $t_n$, $z_n$ is the numerical solution at time $t_n$, $\langle \ \rangle$ means ensemble average (expectation value), $f(z)$ is a four times differentiable function of at most polynomial growth, and $\gamma$ is some constant (Kloeden and Platen, 1992, p. 327).

If a method convergences in the weak sense, this implies that the moments of the modelled distribution converges to the moments of the true distribution, as $\Delta t \to 0$. Hence, a weakly convergent method means that the distribution of numerical

solutions will converge to the true distribution, which means that we can use the method to predict concentrations. There exists methods which have higher order of weak convergence, but we have chosen to consider Euler-Maruyama as it is commonly seen in the literature. See, *e.g.*, Gräwe (2011); Gräwe et al. (2012) for reviews of higher-order methods in the context of water column models.

### A2.2  Reaction terms

As described for the Eulerian implementation in Section A1.4, we consider only one type of reaction term in this paper. This consists of adding mass in certain regions of the domain. Adding particles is done in a stochastic manner, designed to be consistent with a source term in the Eulerian description. For a source term that adds mass at some rate, in some region of the domain, we add a random number of particles at each timestep, such that the expectation value of mass added at each timestep is equal to the integrated source term. The position of each added particle is drawn from a distribution that corresponds to





the spatial distribution of the source term. As this applies only to one of the three cases considered, the details are given in
Section 4.3.

### A2.3   Obtaining a concentration profile from a distribution of particles

Mathematically, the link between the Lagrangian and the Eulerian methods is that each solution of the SDE in the Lagrangian
method represents a sample from the distribution evolving under the advection-diffusion equation in the Eulerian method.
However, in practical applications we are often interested in the distribution itself, and not just samples. Numerous methods
exist for reconstructing the distribution from samples, as this is a common problem not just in applied geoscience, but in
statistics in general. For a review of a few different approaches in the context of applied oceanography, see, *e.g.*, Lynch et al.
(2014, Chapter 8).

Here, we have chosen to use the simple approach of constructing a histogram of particle positions. This works well when
the number of particles is large relative to the number of cells, as is the case for our 1D examples below. An additional benefit
is that the only free parameter is the bin size of the histogram, and a very direct comparison between Lagrangian and Eulerian
results is obtained by setting the bin size equal to the cell size of the Eulerian approach.

### Appendix B:  Distribution of terminal velocities for microplastics

To obtain a distribution of rising/settling velocities for microplastic particles, we combine the distributions for density, size
and shape presented by Kooi and Koelmans (2019) with the empirical relations for terminal velocity obtained by Waldschläger
and Schüttrumpf (2019). In the following, we describe the steps leading to a full distribution of rising/settling velocities. The
Python code used to obtain the distribution is available on GitHub, both as a script and as a Jupyter Notebook [2].

We employ a Monte Carlo approach to obtain a velocity distribution, where we draw a sample from each of the distributions
describing density, size and shape, which we can then map to a terminal velocity via the relations of Waldschläger and Schüt-
trumpf (2019). By repeating this process a large number of times, we obtain a distribution of samples of rising and settling
speeds, which are consistent with the underlying distributions of density, size and shape. In the Lagrangian implementation, we
can use the sampled velocities directly, while in the Eulerian implementation we draw a very large number ($10^9$) of samples,
and map them to discrete classes by creating a histogram with the required number of bins.

### B1   Density distribution

Kooi and Koelmans (2019) found density to be accurately described by a normal-inverse Gaussian distribution, with the pa-
rameters $\mu = 0.84$, $\delta = 0.097$, $\alpha = 75.1$, and $\beta = 71.3$. Generating random samples from this distribution is straightforward,
for example using the class `scipy.stats.norminvgauss` from the SciPy library.

---

[2]https://github.com/nordam/Eulerian-and-Lagrangian-methods





## B2 Size distribution

The size distribution found by Kooi and Koelmans (2019) is a truncated power law distribution for the number of particles,
where the probability number density for particle size, $s$, in micrometers, is

$$p_N(s) = P_0 s^{-\alpha} \tag{B1}$$

with $\alpha = 1.6$, where $s$ is the size in micrometers, and where the range of the distribution is from $20\,\mu\text{m}$ to $5\,\text{mm}$. $P_0$ is a normalisation factor given by

$$P_0 = \left( \int\limits_{20\,\mu\text{m}}^{5\,\text{mm}} s^{-1.6}\,\text{d}s \right)^{-1} \approx 3.7573144455. \tag{B2}$$

To draw random samples from the distribution with probability density function (PDF) given by Eq. (B1), we use the inversion method (see, *e.g.*, Devroye (1986, pp. 27–28)). We let

$$F(s) = \int\limits_{20\,\mu\text{m}}^{s} p_N(s')\,\text{d}s' \tag{B3}$$

be the cumulative distribution function (CDF) of the size distribution. Then, random samples from this distribution can be generated by evaluating $F^{-1}(U)$, where $F^{-1}$ is the inverse of $F(s)$ (which can be found analytically in this case), and $U \in [0,1]$
is a uniformly distributed random number.

## B3 Shape distribution

Kooi and Koelmans (2019) provide a shape distribution for microplastics in terms of the Corey shape factor (CSF), which is defined as

$$\text{CSF} = \frac{H}{\sqrt{LW}}, \tag{B4}$$

where $L$, $W$, and $H$ are respectively the length, width and height of a particle. We assume that $L \geq W \geq H$, and that the values are scaled by $L$ such that $L = 1$ and $W, H \in [0,1]$. While Kooi and Koelmans (2019) obtain a single distribution for CSF in the end, the underlying data is presented as distributions for the width and height for different shape categories, with different relative abundances: fibres (48.5%), fragments (31%), beads (6.5%), films (5.5%), and foam (3.5%). Note that these relative abundances only add up to 95%, hence we have normalised them by dividing each one by 0.95, and we thus achieve
the probabilities listed in Table B1.

For each of the shape categories, the distributions for width and height are assumed to be symmetric, triangular distributions, as described in Kooi and Koelmans (2019), and the distribution parameters for each shape are given in Table S2 of the supplementary materials of Kooi and Koelmans (2019). We also list the distribution parameters in Table B1 for completeness. Generating random samples from a triangular distribution is straightforward, for example with `scipy.stats.triang` from
the SciPy library.



**Table B1.** Probability for a microplastic particle to belong to different shape categories, as well as the parameters of the symmetric, triangular distributions for width and height for each shape category (Kooi and Koelmans, 2019).

| Shape | Fibre | Fragment | Bead | Film | Foam |
|---|---|---|---|---|---|
| Probability | 0.51 | 0.33 | 0.068 | 0.056 | 0.036 |
| Width, low | 0.001 | 0.1 | 0.6 | 0.1 | 0.1 |
| Width, high | 0.5 | 1.0 | 1.0 | 1.0 | 1.0 |
| Height, low | 0.001 | 0.01 | 0.36 | 0.001 | 0.01 |
| Height, high | 0.5 | 1.0 | 1.0 | 0.1 | 1.0 |

Based on the above, our algorithm for the generation of a random CSF for a microplastic particle is as follows:

1. Select a shape category, with the probabilities from Table B1.

2. Generate a random width, $W$, and height, $H$, by drawing from the triangular distributions for that shape category.

3. Calculate the CSF from Eq. (B4), using the randomly selected $W$ and $H$, and that $L = 1$ by definition.

**B4 Terminal velocities**

Waldschläger and Schüttrumpf (2019) provide relations for the rising and settling speeds of microplastics particles, in terms of their density, size and shape. Specifically, they give the terminal velocity as

$$v = \sqrt{\frac{4}{3}\frac{d_{eq}}{C_D}\left|\frac{\rho_s - \rho}{\rho}\right|g}, \tag{B5}$$

where $d_{eq}$ is the equivalent diameter (*i.e.*, the diameter of a sphere with the same volume as the particle), $\rho_s$ and $\rho$ are respec-
tively the densities of the particle and the ambient water, $g$ is the acceleration of gravity, and $C_D$ is the drag coefficient.

Four different expressions are given for the drag coefficient, distinguishing between particles that are lighter or denser than water, as well as distinguishing fibres from other shapes ("pellets and fragments"). The expressions are, for sinking non-fibre particles:

$$C_D = \frac{3}{\text{CSF} \times \sqrt[3]{\text{Re}}}, \tag{B6}$$

for rising non-fibre particles:

$$C_D = \left(\frac{20}{\text{Re}} + \frac{10}{\sqrt{\text{Re}}} + \sqrt{1.195 - \text{CSF}}\right) \times \frac{6}{P}^{1-\text{CSF}}, \tag{B7}$$

for sinking fibres:

$$C_D = \frac{4.7}{\sqrt{\text{Re}}} + \sqrt{\text{CSF}}, \tag{B8}$$





and for rising fibres:

$$C_D = \frac{10}{\sqrt{\mathrm{Re}}} + \sqrt{\mathrm{CSF}}. \tag{B9}$$

The drag coefficients are given in terms of the CSF, the Reynolds number, and in one case the Powers' roundness, with the Reynolds number given by

$$\mathrm{Re} = \frac{v d_{eq}}{\nu}, \tag{B10}$$

where $\nu$ is the kinematic viscosity of the ambient fluid. Given that the drag coefficients depend on the Reynolds number, which in turn depends on the velocity, Eq. (B5) was solved numerically by an iterative approach to find the velocity.

The Powers' roundness, $P$, is a set of six classes from 1 (very angular) to 6 (well-rounded), where particles are assigned a class by comparison with a published photograph of reference particles (Powers, 1953). Powers' roundness is not addressed by Kooi and Koelmans (2019), but Waldschläger and Schüttrumpf (2019, Fig. S6, Supplementary materials) show a histogram of the Powers' roundness distribution. Based on this, we have chosen to use a uniformly distributed random integer between 1 and 6 to represent Powers' roundness. Note that the Powers' roundness only appears in the expression for the drag coefficient for rising pellets and fragments, and is thus not used for fibres.

Even though Waldschläger and Schüttrumpf (2019) provide separate drag coefficients for rising and sinking particles, we have chosen to use the same coefficients in both cases. Our reasoning is two-fold: First, from a physical point of view it is expected that two particles with density $\pm\Delta\rho$ relative to water should have terminal velocities with opposite direction but equal magnitude (certainly as long as $|\Delta\rho| \ll \rho$). Second, the drag coefficient of sinking non-fibre particles (pellets and fragments) provided by Waldschläger and Schüttrumpf (2019) scales as $\mathrm{Re}^{-1/3}$, while Stokes' law is

$$C_D = \frac{24}{\mathrm{Re}}. \tag{B11}$$

For small particles, in the lower part of the microplastics size range[3], the sinking velocities predicted with Eq. (B6) can be 1–2 orders of magnitude higher than those predicted by Stokes' law (even with $\mathrm{CSF} = 1$, as would be the case for spherical particles). The drag coefficient provided for rising non-fibre particles, on the other hand, has the same scaling as Stokes' law for small Reynolds numbers. Hence, we have chosen to use this drag coefficient for both rising and sinking particles. In the case of fibres, Eqs. (B9) and (B8) give very similar velocities, but for reasons of symmetry we have chosen to use Eq. (B9) for both rising and sinking fibres.

**B5  Terminal velocity distribution**

Based on the description above, we generate random realisations of particle properties, where each particle has the properties shown in Table B2. From these properties, we can calculate terminal velocities, using the expressions found by Waldschläger and Schüttrumpf (2019), as described above.

---

[3]Note that the expressions given by Waldschläger and Schüttrumpf (2019) were found by fitting to experimentally measured terminal velocities, but as the smallest sinking non-fibre particle in their experiments was 1 mm, the expression is not necessarily valid for much smaller particles.



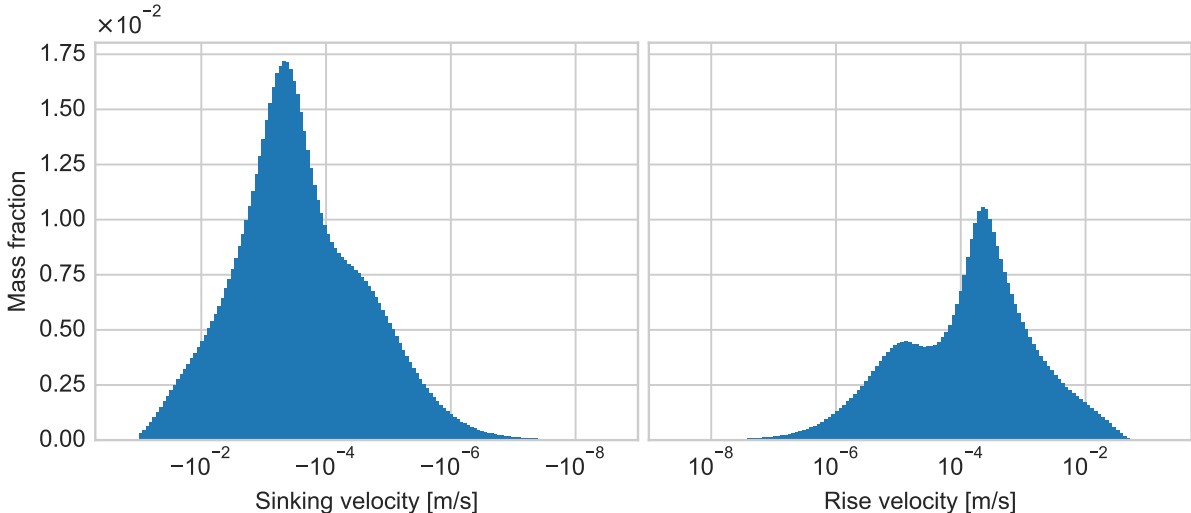

**Figure B1.** Number distribution of terminal particle velocity, found by mapping $10^9$ velocities onto 128 log-spaced negative bins, and 128 log-spaced positive bins, as described in the text.

For the Lagrangian implementation, $N_p$ random velocities were generated at the start of each simulation. For the Eulerian implementation, $10^9$ random velocities were generated, and mapped to $N_k$ different classes. Given the wide range of velocities, we chose to use equally spaced velocity bins on a log scale, with positive and negative velocities treated separately. For positive (rising) velocities, we used bins ranging from $10^{-8}\,\mathrm{m/s}$ to $0.1\,\mathrm{m/s}$, and for the negative velocities we used bins ranging from $-0.3\,\mathrm{m/s}$ to $-10^{-8}\,\mathrm{m/s}$. We used the same number of classes to represent the positive and the negative velocities. By always choosing total number of classes, $N_k$, which is divisible by two, we get a consistent and automated way to divide the velocity distribution into discrete classes. Note that the approach described here was chosen for simplicity and consistency, as our goal is to investigate the numerical convergence with number of classes.

To get the discrete velocity classes, we generated $10^9$ random velocities, and mapped them onto the bins described below, with the velocity of each bin being represented by its midpoint on the logarithmic scale, as described in Section 3.1.2. None of the randomly generated velocities were smaller than $-0.3\,\mathrm{m/s}$ or greater than $0.1\,\mathrm{m/s}$. Approximately 0.02% of the generated velocities fell between $-10^{-8}\,\mathrm{m/s}$ and $10^{-8}\,\mathrm{m/s}$. These were ignored. The bin counts were then converted to normalised mass fractions, such as the total mass sums to 1. Generating $10^9$ random terminal velocities in this manner took approximately 15 minutes on a fairly standard PC.

An example histogram showing the distribution of $10^9$ terminal velocities is shown in Fig. B1, using the same 256 bins that were used for the most accurate Eulerian velocity discretisation. Note that this is a *number* distribution, giving the number of particles with a particular speed. In many cases, a volume (or mass) distribution may be more useful, as this would give the volume of microplastics present as particles with a particular speed. However, obtaining a joint volume probability distribution for size and shape from the number distributions presented by Kooi and Koelmans (2019) is outside the scope of this study.





**Table B2.** Particle properties used to calculate settling speed. All parameters are generated randomly, except equivalent diameter and shape factors, which are calculated from the random parameters as described. Note that the ranges of width and height depend on length, since we assume $L \geq W \geq H$.

| Parameter | Notes |
|---|---|
| Shape category | Fibre or Non-fibre (fragment, bead, film, foam) |
| Density, $\rho$ | Between $0.8\,\mathrm{kg/L}$ and $1.8\,\mathrm{kg/L}$ for 99% of particles |
| Length, $L$ | Between $20\,\mu\mathrm{m}$ and $5\,\mathrm{mm}$ |
| Width, $W$ | Between $0.001L$ and $L$ |
| Height, $H$ | Between $0.001L$ and $W$ |
| Powers roundness, $P$ | Between 0.5 and 6, uniform random distribution |
| Equivalent diameter, $d$ | Between $L$ and $H$, calculated from $d = \sqrt[3]{LWH}$ |
| Shape factor, CSF | Between 0 and 1, calculated from $\mathrm{CSF} = H/\sqrt{LW}$ |

## Appendix C: Additional convergence results

Here we present convergence of the first moment of the distribution with respect to timestep for the Lagrangian implementation, and with respect to timestep and grid cell size for the Eulerian implementation. These results were used to choose suitable
numerical parameters for the investigation of convergence with number of classes and number of particles. They also serve as a sanity check, to demonstrate that the numerical schemes behave as expected.

The results are shown in Fig. C1 (Case 1), Fig. C2 (Case 2), and Fig. C3 (Case 3). In each figure, the left panel shows the convergence of first moment with timestep in the Lagrangian implementation. For each timestep, the first moment has been calculated for each value of the timestep $\Delta t$ as the average particle position over 100 simulations each with $3 \times 10^6$ particles.
The error has then been found by using the average particle position across 100 simulations with $3 \times 10^6$ particles, using a shorter timestep of $\Delta t = 2\,\mathrm{s}$. The convergence appears to be first order, which is as expected for the Euler-Maruyama method, as this method has order of convergence 1 in the weak sense.

We note also that the error in the first moment of the particle distribution for a Lagrangian stochastic method has two terms, a discretisation term due to the timestep, and a stochastic term due to the finite number of samples (Pavliotis, 2014, p. 151). The
expected magnitude of the stochastic error term can estimated from the Lindeberg-Lévy theorem, as described in Section 2.6. In the left panels of Figs. C1, C2, and C3, the magnitude of the stochastic sample error is shown as coloured dashed lines, where we have used $N_p = 3 \times 10^8$ samples, and where the variance of the particle positions for each of the times considered have been assumed to be a good approximation for the (unknown) true variance.

For the Eulerian implementation, the convergence with timestep is shown in the middle panels of Figs. C1, C2, and C3,
where the number of classes and cells have been kept fixed at $N_k = 128$ and $N_z = 8000$. Finally, the convergence with spatial discretisation for the Eulerian method is shown in the right panel of Figs. C1, C2, and C3. Here, the timestep and number of




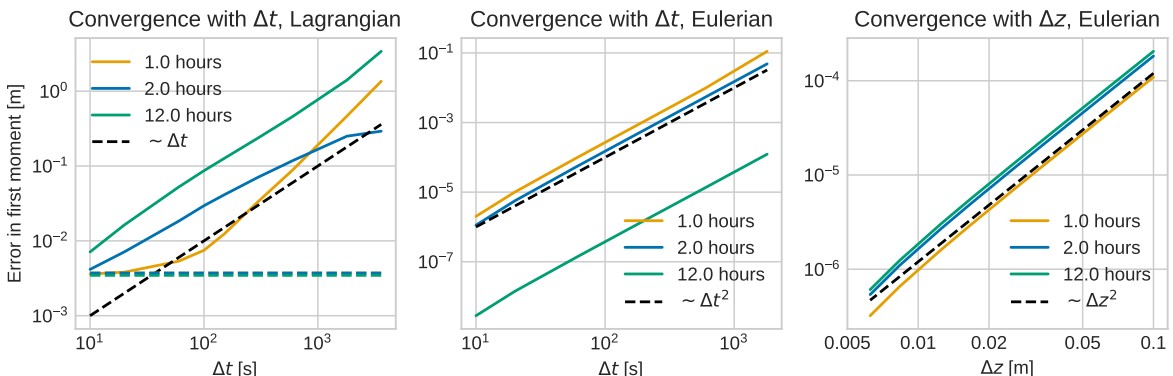

**Figure C1.** Convergence of the first moment of the distribution in Case 1, as a function of timestep for the Lagrangian implementation (left panel), and as a function of timestep (middle panel) and cell size (right panel) for the Eulerian implementation. The coloured dashed lines in the left panel show the level where the sampling error is expected to dominate.

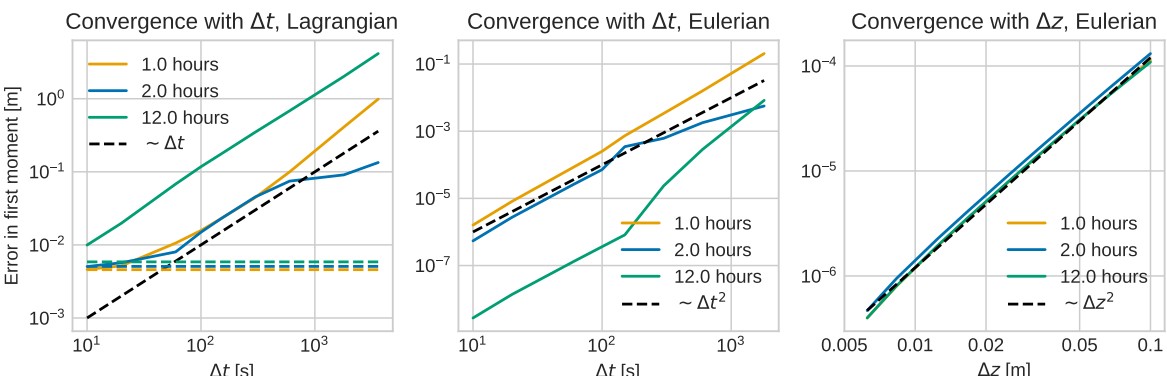

**Figure C2.** Convergence of the first moment of the distribution in Case 2, as a function of timestep for the Lagrangian implementation (left panel), and as a function of timestep (middle panel) and cell size (right panel) for the Eulerian implementation. The coloured dashed lines in the left panel show the level where the sampling error is expected to dominate.

classes have been kept fixed at $\Delta t = 10\,\text{s}$ and $N_k = 128$. We observe that convergence is second order in both space and time, as expected.



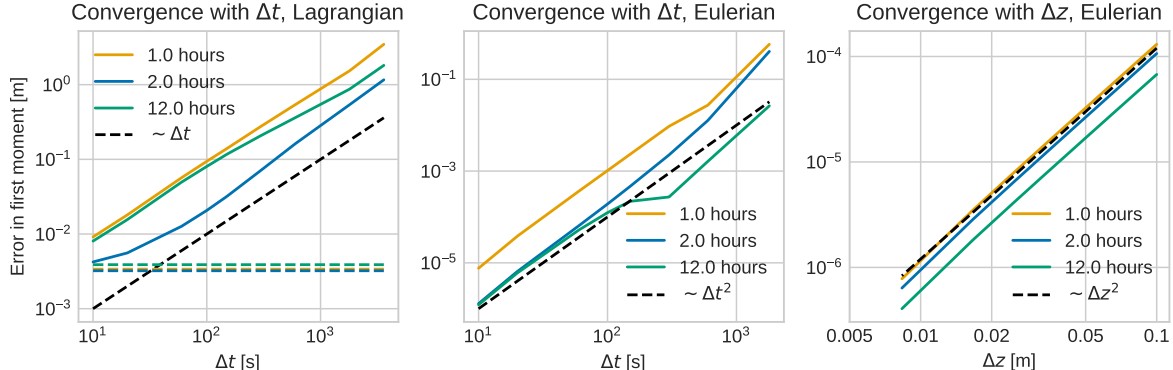

**Figure C3.** Convergence of the first moment of the distribution in Case 3, as a function of timestep for the Lagrangian implementation (left panel), and as a function of timestep (middle panel) and cell size (right panel) for the Eulerian implementation. The coloured dashed lines in the left panel show the level where the sampling error is expected to dominate.

*Author contributions.* TN wrote the Lagrangian implementation, ran all simulations, and wrote the first draft of the manuscript. RK wrote
the Eulerian implementation. All authors contributed significantly to the review and editing of the manuscript.

*Competing interests.* The authors declare that they have no competing interests.

*Acknowledgements.* The work of T.N. was supported in part by the Norwegian Research Council project INDORSE (267793), and in part by
the European Union's Horizon 2020 research and innovation programme under grant agreement no. 814426 – NanoInformaTIX. The work
of E.v.S was supported through funding from the Netherlands Organization for Scientific Research (NWO), Earth and Life Sciences, through
project OCENW.KLEIN.085 and through funding from the European Research Council (ERC) under the European Union's Horizon 2020
research and innovation program (grant agreement 715386). The authors would also like to thank Prof. S. Sundby for helpful advice on the
velocity distribution of pelagic fish eggs.





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
