# Peer review of "A comparison of Eulerian and Lagrangian methods for vertical particle transport in the water column"

_Geoscientific Model Development, 2023_

## Author Comment (AC2)

**Response to reviewers**

Dear Reviewers,

Thank you both for your time, and your comments on our manuscript. Below, we reply to each of the points raised, and explain how we have amended the manuscript in response.

Sincerely,
T. Nordam, R. Kristiansen, R. Nepstad, E. van Sebille, and A. Booth

**Reply to Anonymous Referee #1**

**Comment 1** *In general, I found the analysis rigorous, but the draft organization could be improved. For instance, there are many subsections and continuous cross-references to other sections in the text. The overall result is often a redundancy of notions and a not fluent reading in some parts. I suggest defining and developing the methods and the results separately.*

**Reply:** In response to this, and a similar comment from the second referee, we have made an effort to streamline the manuscript and reduce the amount of cross-referencing.

———————

**Comment 2** *Finally, The relevance of the research questions and the outcomes of the present work are not evident. The fact that Eulerian and Lagrangian Schemes give similar results is not surprising and trivially expected. The implementation of the boundary conditions adopted is also well-established.*

**Reply:** As we discussed in the comments, we believe that boundary conditions for Lagrangian models are not quite as well established as for Eulerian models. We would also say that the equivalence of Eulerian and Lagrangian models is not completely trivial, as there are conditions that must be fulfilled, including conditions on the smoothness of the drift and diffusion coefficients, and of course the aforementioned boundary conditions.

———————

**Comment 3** *Finally, how can one transpose the present results, which are based on a 1D scheme, to more realistic 3D scenarios where the dynamics of the particles are more complex than those implemented here? What is the best approach in the latter cases?*

**Reply:** We would actually argue that the dynamics of the particles is not that much more complex in 3D. It is of course true that there will be horizontal advection and diffusion, but this affects all particles equally, whereas vertical motion is different for particles with different sizes, densities, shapes, etc. Hence, we believe a 1D model will actually capture a large part of the complexity we are trying to address here, *i.e.*, the modelling of particles with different distributions of rising/settling velocities.

Of course one issue in 3D that is less significant in 1D is the need for larger number of particles to accurately represent the distribution of terminal velocities also in areas with dilute concentrations. However, we believe this point is harder to answer generally, as it would depend very much on the case to be modelled, the nature of the horizontal transport, etc. This would have to be addressed in a future work with a full 3D model and realistic forcing data, such as modelled ocean currents.

———————

**Comment 4** *- L210. w instead of v within the square brackets?*

**Reply:** This has been corrected
* * *
**Comment 5** - *L270-279. This long paragraph seems more suited for an Introduction.*

**Reply:** The other reviewer also pointed this out, but suggested moving the text to the discussion. We found that we prefer this to go in the discussion, as it feels a little too technical for the introduction.
* * *
**Comment 6** - *L283. Why do you use the reflection of the particle displacement? Which is the physical meaning? The free surface should dampen the particle velocity independently of the mechanism involved (advection or scattering by diffusivity).*

**Reply:** We believe this question serves to highlight our claim that boundary conditions for Lagrangian particle methods is not quite a well-known issue.

The reason that we use reflection about the boundary is quite simple. In a particle model with a finite (*i.e.*, non-zero) timestep, we need a method to deal with particles that are randomly displaced to a point outside the boundary, and reflection happens to be the simplest approach we are aware of that is consistent with the diffusion equation in the case of constant diffusivity, and appears to converge to the correct solution of the diffusion equation also in the case of non-constant diffusivity. To see this is fairly straightforward: If we have an initial $\delta$-function distribution, constant diffusivity, and a nearby reflecting boundary, then the analytical solution of the diffusion equation is a sum of two Gaussian profiles, placed symmetrically about the boundary. If we simulate this with a random walk scheme using the reflection scheme at the boundary, we get an excellent match with the analytical solution.

It is not quite clear what is meant by "dampen the particle velocity", as a random walk model does not have a variable that represents the diffusive velocity. However, one interpretation might be that the particles should be be stopped at the boundary if they happen to go outside. This can also be described as *projecting* the particles onto the boundary, rather than *reflecting* them about the boundary. If we try this approach on the idealised test case described above, with any non-zero timestep, there will be an excess of particles exactly at the boundary due to the particles that went outside, and this is clearly inconsistent with the analytical solution, as illustrated in Fig. 1.
* * *
**Comment 7** - *Fig.1. The GOTM simulation predicts null diffusivity at the water interface. This seems consistent we the null diffusivity flux at the boundary. Is much more demanding to use the output from GOTM instead of Equation (15) in the transport equation?*

**Reply:** First, we would point out that we believe the zero diffusivity at the boundary in the GOTM results to be an implementation choice in GOTM rather than a true physical prediction. If we run a GOTM simulation with much higher vertical resolution, it appears that the diffusivity approaches a small but non-zero value as we approach the boundary, but then the boundary point is still exactly zero (see Fig. 2 below). In the case considered, where we have assumed a wind speed of 9 m/s, there will be a fair bit of breaking waves, and hence a diffusivity of zero at the surface is perhaps not the best representation of reality.

As to whether it is more demanding to use the output of GOTM directly, we would say that using a diffusivity profile with zero in one or more points at least makes it harder to demonstrate that the problem is well-posed, and that the two different implementations actually model the same thing. For example, in a finite volume approach, such as we have used for our Eulerian implementation, the point exactly at the surface is not resolved, as the first cell starts at the surface and extends downwards as distance $\Delta z$. In a finite difference discretisation, on the other hand, the point at the surface could be resolved, and this point could use a diffusivity value of zero, but that would effectively mean that the region with zero diffusivity extends downwards a distance $\Delta z/2$, and the amount of mass that gets stuck in this region will depend on the resolution.

[Figure]

Figure 1: A simple demonstration of two different boundary schemes in a diffusion simulation using a random walk model, with diffusivity $K = 0.1$, integration time $T = 1$, $N_p = 10000000$ particles initially located at $x = 1$, and a boundary at $x = 0$. To the left: reflection about the boundary. To the right: projection onto the boundary (*i.e.*, particles that go outside are placed exactly at the boundary).

[Figure]

Figure 2: An illustration of predicted diffusivity close to the surface with GOTM, at different vertical resolutions.

For the particle model, positively buoyant particles can perfectly well have a position exactly at the surface, and if the diffusivity in this point is zero, they will remain there forever, at least with the implementation we have used. In reality, for a case like the fish eggs, the eggs are not found to accumulate at the surface, but to be distributed in the water column (see, *e.g.*, Fig. 3 in Sundby (1983), where measured and modelled concentrations of pelagic fish eggs are compared).

`https://www.sciencedirect.com/science/article/abs/pii/0198014983900420`
* * *
**Comment 8** - *L416. Why this difference? Is it maybe owing to a substantial variation of the velocity distribution for the two classes?*

**Reply:** We do indeed believe that the difference between Cases 1 and 2 is due to the different velocity distributions. This was discussed in the discussion, but we will highlight this to make it more clear.
* * *
**Comment 9** - *Fig.4. The graduation of the two y-axes overlaps.*

**Reply:** This has been addressed
* * *
**Comment 10** - *Eq. (19). Which is the value of $H_s$?*

**Reply:** This information has been added to the manuscript in the description of the case.
* * *
**Reply to Alethea Mountford, Referee # 2**

**Comment 11** *Overall, the manuscript presents an interesting discussion on the advantages and disadvantages of each approach in terms of efficacy within a one-dimensional domain, but I wonder how applicable these results would be in a three-dimensional domain.*

**Reply:** As described above, in our reply to comment 3, we believe that our results have some value also for 3D simulations, as a significant part of the challenge in modelling realistic particle distributions come from the distribution of vertical rising or settling speeds.
* * *
**Comment 12** *As mentioned in the general comments, the discussion and description of Eulerian and Lagrangian methods does feel slightly imbalanced in favour of Lagrangian methods. There is only "a challenge" discussed for an Eulerian approach, whereas there is only "an advantage" included for a Lagrangian approach. Both have advantages and disadvantages and are useful for certain applications than the other. A brief mention of the fact that they are quite complementary methods and can be used in conjunction (e.g. to model larval dispersal (Young et al. 2015) and (on a slightly larger scale) to implement transport of interactive icebergs (Marsh et al 2015) to give a couple of examples) would maybe make a good addition to the discussion.*

**Reply:** This is an excellent point, and the additions are good suggestions. We have amended the discussion a bit and added the suggested references, as well as a reference to a paper with a two-way coupled system of an Eulerian and a Lagrangian model for frazil ice.

`https://doi.org/10.3189/2015AoG69A657`
* * *
**Comment 13** *The phrase "implementing different boundary conditions and a simple reaction term" (and variations) is repeated several times in the introduction, and maybe doesn't need to be reiterated quite that frequently.*

**Reply:** This has been addressed.
* * *
**Comment 14** *In Section 2.5, the reader is pointed to both Section A1.4 and Section A2.2 for further details of the reaction term. However, there is not much further discussion of the reaction term aside from that the study only considers reaction terms that add mass, and then points the reader to section 4.3 for more details. This cross-referencing is a little frustrating and would make for an easier read if further detail was provided in one place. I'm not sure that the appendix sections are necessary.*

**Reply:** This has been addressed, and the text from sections A1.4 and A2.2 has been worked into the main text.

———————

**Comment 15** *In Section 3.1.2 L206-208, some more clarity on what "some cases" and "other cases" would be insightful and less vague (are the cases the ones discussed in this study, or just generally?), similarly clarifying what "available directly" means.*

**Reply:** This has been change to make it clear that we are talking about different situtations more generally. A reference with an example of a directly specified velocity distribution has also been added.

———————

**Comment 16** *In Section 3.2.1, the discussion around boundary conditions (L270-278) would maybe better placed in the introduction, appendix or in Section 5.1 where boundary conditions (and the associated challenges) are further discussed.*

**Reply:** This has been moved to the discussion, as suggested.

———————

**Comment 17** *Figures 2, 3 and 5: For all of the left panels of these figures, a clearer distinction between the colours for 0 hours and 2 hours would be very beneficial (and perhaps a less dark colour for 12 hours). Although it is clear which is which from the description of the initial Gaussian vertical distribution of the particles, I think it would be more accessible/easier to understand for those who are perhaps not so familiar with the topic (or those who may have not thoroughly read the text and are heading straight for the figures).*

**Reply:** We will remake these figures with a new colour scheme

———————

.

**Comment 18** *L507: The Lindeberg-Lévy CLT acronym should be defined either earlier on in the text when it is mentioned previously or here.*

**Reply:** Thanks for pointing this out. We have changed this to simply "the Lindeberg-Lévy theorem", as that is the term used in the reference provided.

———————

**Comment 19** *L28: largel -> large*

**Reply:** This has been addressed

———————

**Comment 20** *L32: correct reference for the NEMO manual is Madec et al. (2022), this also needs addressing in the references*

**Reply:** This has been addressed, there was a missing comma in the bibtex file causing Madec, Gurvan to become Madec Gurvan.

———————

**Comment 21** *L186/L255: "we wish to use" should perhaps be reworded as a more definite statement – you have already used the boundary conditions*

**Reply:** This has been changed to simply "we use".

———————

**Comment 22** *L210: should it be w (not v) in the square brackets?*

**Reply:** Yes, this has been addressed

––––––––––

**Comment 23** *L273: particle -> particles*

**Reply:** This has been addressed

––––––––––

**Comment 24** *L275: missing closing bracket*

**Reply:** This has been addressed

––––––––––

**Comment 25** *L556: ... both broad and changing time ... -> changing over time*

**Reply:** This has been addressed

––––––––––